# 🦉OWL: Geometry-Aware Spatial Reasoning for Audio Large Language Models

**Subrata Biswas**[*], **Mohammad Nur Hosssain Khan**[*] **& Bashima Islam**
Department of Electrical & Computer Engineering
Worcester Polytechnic Institute
Worcester, MA 01609, USA
{sbiswas, mkhan, bislam}@wpi.edu

## Abstract

Spatial reasoning is fundamental to auditory perception, yet current audio large language models (ALLMs) largely rely on unstructured binaural cues and single-step inference. This limits both perceptual accuracy in direction and distance estimation and the capacity for interpretable reasoning. Recent work such as BAT demonstrates spatial QA with binaural audio, but its reliance on coarse categorical labels (left, right, up, down) and the absence of explicit geometric supervision constrain resolution and robustness. We introduce the **Spatial-Acoustic Geometry Encoder (SAGE)**, a geometry-aware audio encoder that aligns binaural acoustic features with 3D spatial structure using panoramic depth images and simulated room-impulse responses at training time, while requiring only audio at inference. Building on this representation, we present **OWL**, an ALLM that integrates **SAGE** with a spatially grounded chain-of-thought to rationalize over direction-of-arrivals (DoA) and distance estimates. Through curriculum learning from perceptual QA to multi-step reasoning, **OWL** supports o'clock-level azimuth and DoA estimation. To enable large-scale training and evaluation, we construct and release **BiDepth**, a dataset of over one million QA pairs combining binaural audio with panoramic depth images and room impulse responses across both in-room and out-of-room scenarios. Across two benchmark datasets, our new **BiDepth** and the public SpatialSoundQA, **OWL** reduces mean DoA error by **11°** through **SAGE** and improves spatial reasoning QA accuracy by up to **25%** over BAT. Our dataset and code are available at: https://github.com/BASHLab/OWL

## 1 Introduction

Large language models (LLMs) Achiam et al. (2023); Team et al. (2023); Touvron et al. (2023) have catalyzed rapid progress beyond pure text, inspiring multimodal systems Biswas et al. (2025) that pair an LLM backbone with modality encoders for vision Liu et al. (2023); Li et al. (2023), audio Gong et al. (2023); Ghosh et al. (2025); Goel et al. (2025), and other sensors Imran et al. (2024); Leng et al. (2024); Ouyang & Srivastava (2024). Through projection layers for cross-modal alignment, these systems can process heterogeneous inputs, reason over joint representations, and follow instructions in context by training on paired *(modality, text)* data. Within audio, such models learn to align acoustic features with language to parse complex sound scenes, recognize events and speaker attributes, and carry out dialogue conditioned on what they *"hear."* Early results show strong zero-shot generalization and retrieval capabilities when trained at scale, and instruction tuning further enables conversational audio-conditioned queries.

Despite this momentum, audio-augmented multimodal LLMs lag behind their vision-language counterparts Liu et al. (2024); Bai et al. (2023); Li et al. (2024) due to the unique challenges of sound: long-range temporal dependencies, nonstationary noise, and the need to capture geometric cues of the acoustic environment that shape auditory perception. Even advanced models such as `Gemini-2.5-flash` Comanici et al. (2025) struggle with composite acoustic tasks that require fine-grained spatial reasoning, highlighting persistent gaps in audio-language understanding. Recent

---

[*] Equal Contribution.

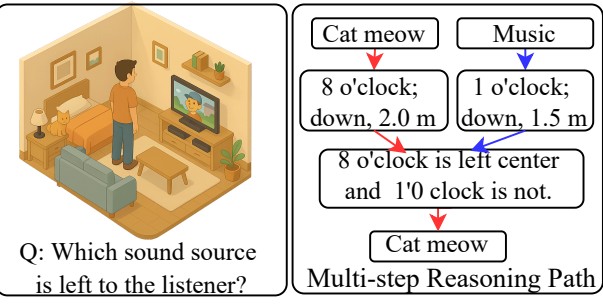

Figure 1: **SAGE** encodes binaural audio into spatially grounded representations. **OWL** detects events, localizes by direction and distance, and applies multi-step reasoning, yielding interpretable rationales for queries such as *"Which sound source is left of the listener?"*

advancements, such as BAT Zheng et al. (2024b), demonstrates spatial reasoning capabilities, but its localization remains coarse, subdividing the scene into only four broad regions (e.g., front-left, left-behind). However, many downstream tasks, such as fine-grained source tracking, relative distance estimation, and multi-source disambiguation, require a more precise understanding of spatial cues.

These gaps stem from two fundamental limitations in current audio large language models (ALLMs). *(i) Lack of geometric grounding.* Existing encoders capture spectral and temporal patterns but overlook crucial spatial cues such as direct-to-reverberation ratios, reverberation time (RT60), and room layout that determine how sound propagates. As a result, models can recognize sound events but fail at spatial reasoning tasks such as deciding which source is closer or determining whether a sound originates from the left or right. *(ii) Single-pass reasoning.* Current ALLMs map questions directly to answers without intermediate inference steps. This prevents them from decomposing complex acoustic queries into smaller, interpretable subproblems. Consequently, they falter in multi-source scenes and queries that require step-by-step spatial reasoning.

We address these limitations by proposing **OWL**, a framework powered by the **S**patial-**A**coustic **G**eometry **E**ncoder (**SAGE**). Unlike BAT, **OWL** combines geometry-conditioned training with a spatially grounded chain-of-thought (CoT), enabling finer localization and structured inference for complex queries. **SAGE** jointly models acoustic and geometric properties of environments, incorporating cues such as directionality, distance-dependent reverberation, and room structure. During training, it leverages binaural room impulse responses (RIRs) and paired panoramic depth images to learn how geometry shapes sound, but at inference it requires only binaural audio, making it broadly applicable. While **SAGE** provides geometry-aware acoustic representations, **OWL** extends this with a spatially grounded CoT mechanism that anchors intermediate reasoning steps to source locations. Rather than answering in a single pass, **OWL** localizes and interprets spatial configurations, then performs structured inference. This decomposition from perception (localization and detection) to reasoning leads to more accurate and interpretable responses, for example, generating rationales such as 'sound A at 8 o'clock is left of sound B at 1 o'clock,' as illustrated in Figure 1.

To support this pipeline, we construct **BiDepth**, a large-scale public dataset that couples binaural audio, binaural RIRs, panoramic depth images, and question-answer annotations. Unlike prior datasets such as SoundScape Pano-IR and SpatialSoundQA, **BiDepth** integrates all three modalities to provide explicit geometric supervision for training **SAGE** and **OWL**. It contains over **1.1M** questions spanning perceptual QA, spatial reasoning, and CoT-augmented multi-step QA. To reduce template bias and leakage, **BiDepth** includes linguistic variants and a split design that prevents overlap in rooms and sources between train and test.

We validate our approach on both standard SELD tasks and new spatial reasoning benchmarks. Results show that **SAGE** consistently outperforms state-of-the-art (SOTA) methods in sound event localization and detection (SELD) Adavanne et al. (2018), achieving a 1.71% gain in mean average precision, an 11° reduction in mean angular error, and a 33.5% decrease in distance error rate. **OWL** surpasses BAT by 46.4% on perceptual QA and 24.9% on spatial reasoning benchmarks, using a sparse variant aligned with BAT's coarse categories; our full model further supports fine-grained 12-sector DoA estimation and spatial reasoning. Our contributions can be summarized as follows:

• **BiDepth**, the first large-scale dataset ($\approx$1.1M QA pairs) of {binaural audio, binaural RIR, depth image, QA} 4-tuples with geometric grounding for perception and multi-step spatial reasoning.

• **SAGE**, A novel spatially grounded acoustic encoder trained with multimodal supervision, requiring only audio at inference for efficient, geometry-aware deployment.

• **OWL**, a spatial ALLM integrating **SAGE** with spatially grounded CoT reasoning, unifying event detection, localization, and structured inference to achieve SOTA SELD and spatial QA performance.

## 2 RELATED WORKS

**Audio Large Language Models**. Contrastive audio–language pretraining (e.g., CLAP Wu et al. (2023)) laid the foundation for retrieval and zero-shot transfer. Subsequent audio LLMs Deshmukh et al. (2023); Gong et al. (2023); Huang et al. (2024); Tang et al. (2023); Chu et al. (2023); Biswas et al. (2025) scaled datasets and training strategies but remained perception-oriented, focusing on recognition or QA. The Audio Flamingo family Kong et al. (2024); Ghosh et al. (2025); Goel et al. (2025) further broadened task coverage, enhancing performance across diverse applications. Yet, most models still emphasize perception-oriented classification and QA while overlooking spatial reasoning, which is essential for progress toward audio general intelligence Morris et al. (2023). BAT Zheng et al. (2024b) is a notable exception, introducing spatial QA from binaural audio, but it reduces scenes to coarse bins (front, back, left, right), uses single-step inference, and lacks geometric grounding since its encoder is trained with audio alone. In contrast, our approach leverages geometry-conditioned training to align audio with spatial structure for better localization and multi-step reasoning.

**CoT Reasoning in Multimodal LLMs**. Chain-of-thought (CoT) reasoning has been widely adopted to improve stepwise inference in multimodal LLMs. Prior work has exploited graphical cues from images Deng et al. (2024); He et al. (2024); Thawakar et al. (2025), logical structures Dong et al. (2025); Xiao et al. (2024); Zheng et al. (2024a), and textual prompts Bi et al. (2024); Xu et al. (2024); Chen et al. (2024b), achieving stronger interpretability and accuracy in vision-language tasks. In audio, however, CoT is largely absent: to our knowledge, Audio Flamingo 3 Goel et al. (2025) is the only prior attempt and is limited to simple perceptual queries without spatial grounding. Our work fills this gap by introducing geometry-aware CoT for audio, anchoring intermediate reasoning steps to source locations and enabling fine-grained 3D acoustic reasoning.

**Sound Event Detection and Localization**. Sound event detection and localization (SELD) jointly addresses sound event recognition and direction-of-arrival (DoA) estimation. The DCASE challenge standardized this task, with baselines such as SELDnet Adavanne et al. (2018) and the ACCDOA formulation Shimada et al. (2021) widely adopted. Later architectures (CRNNs Biswas et al. (2023), Conformers Gulati et al. (2020), GRUs Cho et al. (2014)) further improved accuracy. However, SELD methods remain task-specific, rely solely on audio features, and lack explicit geometric grounding. In contrast, our approach leverages an LLM framework that aligns audio with environmental geometry and extends beyond detection and localization to support interpretable, multi-step spatial reasoning.

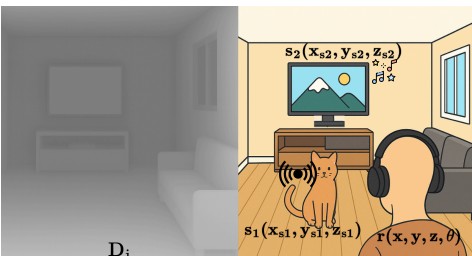

Figure 2: Example of paired modalities in **BiDepth**. **Left:** panoramic depth image $D_i$ capturing geometric context from the listener's perspective. **Right:** binaural acoustic simulation, we place sound sources at positions $s_1$ and $s_2$ and place the receiver at location $r$. For illustration, a cat sound can be positioned at $s_1$.

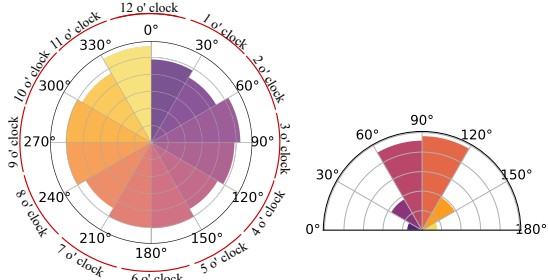

(a) Azimuth distribution  (b) Elevation distribution

Figure 3: Azimuth and elevation angle distributions in **BiDepth**, , showing source directions relative to the listener. Azimuths are nearly uniform, while elevations cluster near the horizontal plane.

## 3 BIDEPTH DATASET

Audio QA datasets (e.g., SpatialSoundQA Zheng et al. (2024b), SpatialVLM Chen et al. (2024a)) include binaural RIRs but lack geometric cues, unlike vision-language QA datasets (e.g., CLEVR Johnson et al. (2017), GQA Hudson & Manning (2019)) that use depth for relational reasoning. We introduce **BiDepth**, a synthetic dataset that couples binaural RIRs with panoramic depth and a CoT QA corpus with stepwise rationales, comprising 28,000 RIR-depth pairs and 1.1 million QA 4-tuples.

### 3.1 ACOUSTIC-GEOMETRIC SIMULATION

We adopt a simulation-based approach to generate paired acoustic and geometric data, enabling systematic variation across environments with scalability and reproducibility. Unlike real-world measurements that require specialized hardware and cover limited spaces, simulation offers controlled access to diverse layouts, surface materials, and source-receiver configurations. Our dataset is built with `SoundSpaces v2.0` Chen et al. (2020; 2022) and `Matterport3D` Chang et al. (2017) (90 buildings, $\approx 24$ rooms per building, 30 scene types). For each RIR, a binaural receiver is placed at $r = (x, y, z, \theta)$, with random location $(x, y, z)$ and orientation $(\theta)$. A sound source $s$ is uniformly sampled within 10 m of the receiver $r$ to remain inside the building. The RIR encodes the acoustic transfer function from $s$ to $r$, capturing spatial and reverberant properties of the environment. Figure 2 shows the simulation setup, including the sound source, binaural receiver, and the panoramic depth image $D_i$ capturing room geometry. Let $M^s(t)$ denote a monaural input signal emitted from the source at $s$. The corresponding binaural signal received at $r$ is then given by

$$B^r(t) = \begin{bmatrix} B^r_L(t) \\ B^r_R(t) \end{bmatrix} = \begin{bmatrix} \text{RIR}_L(t, s, r, \gamma) \\ \text{RIR}_R(t, s, r, \gamma) \end{bmatrix} \circledast M^s(t); \quad t \in [1, T] \text{ and } \circledast \text{ denotes convolution} \quad (1)$$

where $B^r_L(t)$ and $B^r_R(t)$ are the left and right binaural channels, $\text{RIR}_n(t; s, r, \gamma)$ denotes the room impulse response between the source at $s$ and the receiver at $r$ for channel $n \in \{L, R\}$, and $\gamma$ represents the environmental configuration, including geometry, construction materials, and furniture layout. To complement acoustic signals, we render panoramic depth maps from the receiver. Rotating the receiver in $20°$ increments yields depth images $\mathbf{D_i}$ with limited fields of view; concatenation produces a panoramic map encoding walls, obstacles, and room structure. This pairing provides explicit alignment between acoustic propagation and 3D geometry.

With the simulated setup illustrated in Figure 2, we generate 28K unique RIR, depth pairs spanning diverse azimuths, elevations, and distances shown in Figure 3. Unlike prior datasets (e.g., Spatial-SoundQA, SpatialVLM) that provide RIRs without geometry, our dataset explicitly couples acoustics with depth, offering the first large-scale resource for geometry-aware audio reasoning. Representative examples, full rendering details, sampling distributions, and statistics are in Appendix A.

### 3.2 QUESTION-ANSWER GENERATION

We construct QA pairs that integrate auditory cues and geometry for spatial understanding. Following GAMA Ghosh et al. (2024) and LTU Gong et al. (2023), each entry is a quadruplet binaural audio, binaural RIR, depth image, QA, where audio, RIR, and depth provide multimodal supervision while QA drives perception and reasoning. At inference, models receive only audio, with extra signals used for geometry-aware training. BiDepth contains over 1.1M QA pairs balanced across four categories described below (examples of templates and stepwise rationales are in Appendix A.4).

**Type I: Event Detection**. This task identifies the sound sources present in the scene. We include both single-source cases and dual concurrent-source cases, using monaural clips ($M^s(t)$) from AudioSet Gemmeke et al. (2017) spatialized with simulated RIRs.

**Type II: Direction Estimation**. This task estimates the azimuth, elevation, and distance of sources, requiring models to predict direction and distance either from source to receiver or between sources. The horizontal plane is divided into 12 clock-based sectors, elevation is labeled up or down, and distance is expressed in conversational form (e.g., 3 o'clock; up; 2.5 m'), quantized in 0.5 m steps up to 10 m to reflect human approximation.

**Type III: Spatial Reasoning**. This task targets spatial reasoning through relational queries involving two sources and the receiver, such as *'Is source 1 left of source 2?'* or *'Is source 1 closer to the receiver*

*than source 2?'* Formulated as binary *Yes/No* tasks, these queries move beyond absolute localization to assess whether models can reason over relative spatial relationships in complex acoustic scenes.

**Type IV: CoT for Spatial Reasoning**. This task extends spatial reasoning by providing CoT rationales. Instead of binary Yes/No labels, answers include concise reasoning steps. For example, for the query "`Is <source 1> closer to the receiver than <source 2>?`", the rationale may state: *source 1 is 5.0 m away, source 2 is 3.5 m away, therefore source 2 is closer*, yielding the answer No. These rationales make the reasoning process explicit and encourage models to ground predictions in structured spatial comparisons rather than direct classification.

# 4 SAGE: SPATIAL-ACOUSTIC GEOMETRY ENCODER

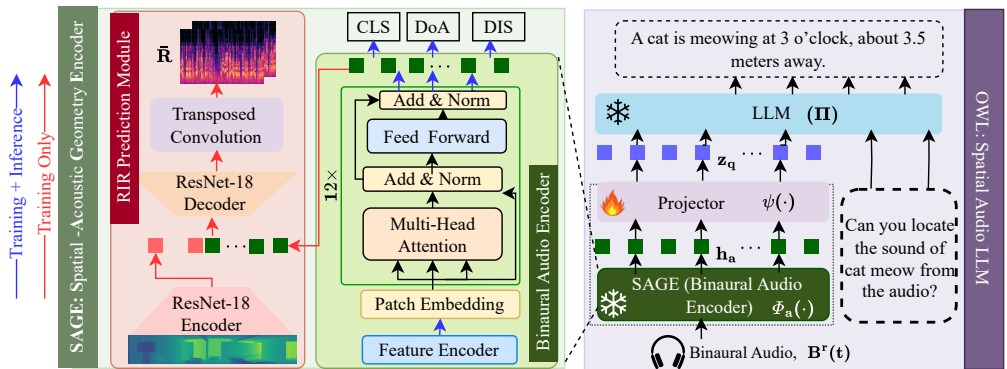

Figure 4: **Architecture of OWL and SAGE.** The left panel shows **SAGE**, trained with geometry-aware supervision using RIRs and depth cues. The right panel illustrates the **OWL** pipeline, where the Binaural Audio Encoder $\phi_{\mathbf{a}}(\cdot)$ is combined with the LLM $\Pi$ through a projector $\psi(\cdot)$ to generate spatially grounded answers. Here, 🔥 and ❄ represent trainable and frozen components, respectively.

We introduce **SAGE**, the first geometry-aware encoder that jointly models acoustic signals and scene structure. The key idea is to inject geometric cues through an auxiliary binaural RIR prediction task, providing privileged supervision that regularizes the audio encoder. **SAGE** comprises two jointly optimized modules: *(i) a binaural audio encoder for perceptual tasks*, and *(ii) an RIR prediction module that fuses depth features with audio representations*. At inference time, only the audio encoder is used, ensuring broad applicability without requiring geometric inputs.

**Binaural Audio Encoder**. The binaural audio encoder $\phi_a(\cdot)$ takes a binaural waveform $\mathbf{B} \in \mathbb{R}^{2 \times L}$ as input, where each channel represents a left or right ear signal of length $L$. It outputs an embedding $\mathbf{h}_a = \phi_a(\mathbf{B}) \in \mathbb{R}^{C \times T}$ that captures spatial and semantic cues across $C$ feature channels and $T$ temporal frames. This representation supports three prediction tasks: sound event classification, direction-of-arrival (DoA) estimation, and distance prediction. For DoA, azimuth is discretized into 360 bins ($[0°, 359°]$) at $1°$ resolution and elevation into 180 bins ($[0°, 179°]$), while for distance the range $[0, 10]$ m is uniformly quantized into 21 bins with 0.5 m spacing. The overall training objective is a weighted sum of cross-entropy losses for event classification ($\mathcal{L}_{\text{cls}}$), distance prediction ($\mathcal{L}_{\text{dis}}$), and DoA estimation ($\mathcal{L}_{\text{doa}}$), where the learnable weight coefficients ($\alpha_i$) balance each task's contribution.

$$\mathcal{L}_{\text{binaural}} = \alpha_1 \mathcal{L}_{\text{cls}} + \alpha_2 \mathcal{L}_{\text{dis}} + \alpha_3 \mathcal{L}_{\text{doa}} \tag{2}$$

**RIR prediction module**. A ResNet-18 encoder $\phi_d(\cdot)$ processes the panoramic depth image $\mathbf{D}_i \in \mathbb{R}^{H \times W}$ and produces a latent representation $\mathbf{h}_d = \phi_d(\mathbf{D}_i) \in \mathbb{R}^{C \times T}$. This depth-derived embedding is fused with the audio features $\mathbf{h}_a$ and decoded by a ResNet-18 transposed convolutional head to reconstruct the binaural RIR, $\bar{\mathbf{R}} = \psi_d(\mathbf{h}_d, \mathbf{h}_a)$.

Given the ground-truth RIR $\mathbf{R}$, reconstruction is supervised by a geometric loss that combines an $\ell_1$ term with an Energy Decay Curve (EDC) loss,

$$\mathcal{L}_{\text{geo}} = \|\mathbf{R} - \bar{\mathbf{R}}\|_1 + \lambda \, \mathcal{L}_{\text{EDC}}(\mathbf{R}, \bar{\mathbf{R}}) \tag{3}$$

where $\mathcal{L}_{\text{EDC}}$ measures the mismatch between predicted and reference decay curves computed via Schroeder's backward integration algorithm Schroeder (1965). Unlike scalar descriptors such as

RT60, the EDC loss is differentiable and captures richer reverberant structure, including direct-to-reverberant ratio (DRR) and early decay time (EDT). This auxiliary supervision encourages the encoder to internalize geometry-aware acoustic features that complement the perceptual tasks.

**Overall Training Objective**. Following prior work Zheng et al. (2024b), we use Mel-Spectrograms and Interaural Phase Difference (IPD) as inputs to $\phi_a(\cdot)$ (see Appendix B.1). The overall training objective combines perceptual and geometric terms, $\mathcal{L} = \eta_1 \mathcal{L}_{\text{binaural}} + \eta_2 \mathcal{L}_{\text{geo}}$. Scalar weights ($\eta_1$, $\eta_2$) balance the contributions. Training uses AudioSet Gemmeke et al. (2017) events spatialized with simulated RIRs to enable joint optimization of acoustic and geometric objectives. At inference, depth images are unavailable and only the audio encoder $\phi_a(\cdot)$ is used for downstream tasks, e.g., event classification, DoA estimation, and distance prediction. See Appendix B.2 for further details.

# 5 OWL: SPATIAL AUDIO-LLM WITH CHAIN-OF-THOUGHT REASONING

**OWL** integrates spatial perception and reasoning over binaural audio by coupling a geometry-aware encoder with a large language model (LLM). The overall architecture, illustrated in Figure 4, comprises three components: (i) the binaural audio encoder $\phi_a(\cdot)$ from **SAGE**, which extracts spatially grounded acoustic features from raw waveforms; (ii) a projection module $\psi(\cdot)$ based on Q-Former Li et al. (2023), where $Q$ learnable query tokens perform cross-attentive pooling to align audio features with the LLM embedding space while reducing sequence length; and (iii) a decoder $\Pi(\cdot)$, instantiated as LLaMA-2-7B Touvron et al. (2023), which conditions on both projected tokens and textual prompts to generate task-specific outputs. Formally, $\phi_a(\cdot)$ maps a binaural input $\mathbf{B}^r(t)$ to an embedding $\mathbf{h}_a \in \mathbb{R}^{C \times T}$, $\psi(\cdot)$ projects $\mathbf{h}_a$ into $\mathbf{z}_q \in \mathbb{R}^{Q \times d}$, and $\Pi(\cdot)$ decodes $\mathbf{z}_q$ together with a text prompt $\mathbf{x}_t$ to produce the output sequence $\mathbf{y} = \Pi(\mathbf{z}_q, \mathbf{x}_t)$. This design enables **OWL** to compress high-dimensional acoustic features into semantically aligned tokens and generate outputs that combine spatial perception with interpretable reasoning.

We adopt Q-Former for its selective cross-attentive pooling, which preserves spatial cues more effectively than lightweight linear or MLP adapters. Following BAT, we use LLaMA-2-7B as the language backbone to ensure fair comparison, while our novelty lies in augmenting it with a geometry-aware encoder and a curriculum for explicit spatial reasoning and Chain-of-Thought supervision. LoRA Hu et al. (2022) enables parameter-efficient adaptation, and $\phi_a(\cdot)$ is kept frozen to retain geometry-aware features learned in **SAGE**. To our knowledge, this is the first curriculum-trained spatial audio LLM with explicit geometry-aware CoT supervision.

## 5.1 TRAINING OF OWL WITH CoT SUPERVISION

We train **OWL** using a three-stage curriculum that progresses from perceptual grounding to relational reasoning and finally to explicit CoT supervision. This staged approach ensures that the model first acquires low-level perception skills before being challenged with higher-level geometric reasoning.

**Stage 1: Perceptual Pre-Training**. The model is first trained on Type I-II QA pairs for event detection and direction estimation. Training begins with

Table 1: Multi-stage training: single-source warmup, then dual-source inputs for reasoning.

| Training Stage | Question Type | No. Audio Source | No. of Train Samples |
|---|---|---|---|
| Stage 1 | I, II | Single (warmup) | 270K |
| | | Dual | 270K |
| Stage 2 | III | Dual | 300K |
| Stage 3 | IV | Dual | 250K |

single-source recordings to stabilize event recognition and DoA estimation before introducing dual-source cases, which require disentangling overlapping cues. This stage grounds **OWL** in basic spatial perception and prevents overfitting to relational shortcuts before learning low-level geometry.

**Stage 2: Relative Geometry Pre-Training**. The model is next exposed to Type III QA pairs, which emphasize relational geometry (e.g., left/right or closer/farther) rather than absolute positions. This stage bridges perceptual grounding and CoT supervision by encouraging **OWL** to internalize structured spatial relations between sources and the receiver. Without this intermediate step, the model struggles to generalize from low-level perception to multi-step reasoning.

**Stage 3: CoT Instruction Tuning**. Finally, the model is trained on Type IV QA pairs that provide full Chain-of-Thought (CoT) explanations alongside the final decision. Supervising both the intermediate reasoning steps and the final prediction aligns the model's outputs with interpretable spatial logic.

Table 2: Comparison of **SAGE**, SELDNet, and Spatial-AST on SpatialSoundQA and **BiDepth**. Best and second-best results are in **bold** and underline. Models are trained on their evaluation dataset unless noted otherwise (see footnotes for setups).

| Method | Modality | | SpatialSound-QA (SSQA) | | | | BiDepth | | | |
|---|---|---|---|---|---|---|---|---|---|---|
| | Audio | Depth | mAP ↑ | $ER_{20°}$ ↓ | MAE ↓ | DER ↓ | mAP ↑ | $ER_{20°}$ ↓ | MAE ↓ | DER ↓ |
| SELDNet | ✓ | ✗ | 42.66 | 25.19 | 19.21 | 38.46 | 39.46 | 53.21 | 38.71 | 53.38 |
| Spatial-AST[1] | ✓ | ✗ | **50.03** | 23.89 | **17.94** | 32.54 | 48.97 | 45.29 | 32.99 | 47.82 |
| Spatial-AST[2] | ✓ | ✗ | - | - | - | - | 49.17 | 41.94 | 27.24 | 39.21 |
| **SAGE**[3] | ✓ | ✗ | 49.71 | 26.59 | 23.19 | 33.03 | 49.75 | 36.89 | 26.32 | 17.11 |
| **SAGE**[4] | ✓ | ✗ | 49.94 | **23.67** | 18.26 | 32.61 | - | - | - | - |
| **SAGE**[5] | ✓ | ✓ | 49.93 | 24.71 | 18.47 | **17.84** | **49.81** | **28.13** | **21.67** | **14.32** |

[1] Trained on SpatialSoundQA.  [2] Trained on SpatialSoundQA and fine-tuned on BiDepth.  [3] Trained on BiDepth audio only.  [4] Pre-trained on BiDepth audio only, then fine-tuned on SpatialSoundQA.  [5] Trained of BiDepth audio and depth.

Unlike prior work, which supplies only categorical labels, this stage enforces explicit step-by-step reasoning, yielding more accurate responses accompanied by human-readable justifications.

**Training Loss**. At each stage we minimize the standard auto-regressive cross-entropy loss over the target token sequence. Given a binaural input $\mathbf{B}^r(t)$, a question $\mathbf{x}_t$, and a target output $\mathbf{y} = (y_1, \ldots, y_T)$, the audio encoder $\phi_a(\cdot)$ produces features that are projected by $\psi(\cdot)$ into query tokens $\mathbf{z}_q = \psi(\phi_a(\mathbf{B}^r(t)))$. The language decoder $\Pi(\cdot)$ then conditions on both $\mathbf{x}_t$ and $\mathbf{z}_q$ to predict each token. Across the three stage-specific datasets $\mathcal{D}_{1-2}$, $\mathcal{D}_3$, and $\mathcal{D}_4$, Equation 4 is the unified training objective. During training, $\psi(\cdot)$ is trained from scratch while $\Pi(\cdot)$ is fine-tuned with LoRA, and $\phi_a(\cdot)$ remains frozen. Table 1 summarizes the datasets used across all stages, with ablations in Section 6.3 and hyperparameter details in Appendix C.

$$\mathcal{L}(\phi_a, \psi, \mathbf{\Pi}) = \sum_{s \in \{1,2,3,4\}} \mathbb{E}_{(\mathbf{B}^r(t),q,y) \sim \mathcal{D}_s} \left[ -\sum_{t=1}^{T} \log_{\mathbf{\Pi}}(y_t | y_{<t}), q, z_q = \psi(\phi(\mathbf{B}^r(t))) \right] \quad (4)$$

# 6 EXPERIMENTS

## 6.1 IMPLEMENTATION DETAILS

**Front-end Audio Processing**. Following BAT, we normalize loudness so each 10-s binaural clip has consistent energy, as in AudioMAE Huang et al. (2022). We compute Short-Time Fourier Transforms (window size = 1024, hop size = 320), then obtain two-channel mel-spectrograms (128 mel bins). We also extract sine and cosine encodings of the Interaural phase difference (IPD). The mel-spectrogram provides spectral energy cues essential for event detection, while IPD encodes inter-channel phase differences critical for localization. See Appendix B.1 for details.

**Evaluation Metrics**. For **SAGE**, we evaluate event detection using mean average precision (mAP), DoA estimation using mean angular error (MAE) and error rate where angular error (azimuth and elevation) exceeds $20°$ ($ER_{20°}$), and distance estimation using the Distance Error Rate (DER), which measures predictions deviating from the ground truth by more than 0.5 m. These metrics jointly assess semantic correctness (mAP) and geometric accuracy (DoA, distance), with results reported on both **SpatialSoundQA** Zheng et al. (2024b) and **BiDepth**. For **OWL**, we use the same measures for event detection, DoA, and distance, and additionally report binary accuracy (BA) for spatial reasoning (Type III) and combined detection, direction, and BA for CoT reasoning (Type IV), under both single- and dual-source conditions (Types I–II). We refer to F for details of evaluation metrics.

**Baselines**. We compare **SAGE** against SELDNet Adavanne et al. (2018) and Spatial-AST Zheng et al. (2024b), two established baselines for sound event localization and detection. For **OWL**, we evaluate against both open- and closed-source multimodal LLMs. The open-source baselines include BAT Zheng et al. (2024b), designed specifically for spatial audio reasoning, and general multimodal models such as VideoLLaMA2 Cheng et al. (2024), RAVEN Biswas et al. (2025), and AudioFlamingo2 Ghosh et al. (2025). For closed-source systems, we run Gemini-1.5-Pro, Gemini-2.5-Pro, and Gemini-2.5-Flash under our evaluation setup to benchmark against the latest proprietary models. Together, these baselines cover both task-specific spatial audio systems and broader audio-augmented LLMs, ensuring a fair and comprehensive comparison.

Table 3: Comparison of **OWL** with closed- and open-source baselines on **BiDepth** across four task types: Type I (event detection), Type II (direction estimation), Type III (spatial reasoning), and Type IV (CoT reasoning). **OWL** consistently surpasses prior open-source models, with further gains from CoT supervision. Best results are in **bold**.

| Method | TypeI Detection (mAP)↑ Single Source | Double source | DoA (Acc)↑ Single Source | Double source | TypeII Distance (DER)↓ Single Source | Double source | TypeIII BA↑ | TypeIV Detection↑ | Direction↑ | BA↑ |
|---|---|---|---|---|---|---|---|---|---|---|
| **Closed-source Models[†]** | | | | | | | | | | |
| Gemini1.5Pro | 31.19 | 12.71 | - | - | - | - | - | 11.96 | - | - |
| Gemini2.5Pro | 32.47 | 12.17 | - | - | - | - | - | 12.01 | - | - |
| Gemini2.5Flash | 32.91 | 12.29 | - | - | - | - | - | 12.21 | - | - |
| **Open-source Models** | | | | | | | | | | |
| VideoLLaMA2 | 17.11 | 5.21 | 12.23 | 11.76 | 68.12 | 83.78 | 8.29 | 5.19 | - | - |
| RAVEN | 16.29 | 5.43 | 13.79 | 9.39 | 71.46 | 82.37 | 9.76 | 5.97 | - | - |
| AudioFlamingo2 | 27.59 | 6.73 | 17.74 | 14.17 | 54.62 | 68.91 | 19.54 | 7.59 | - | - |
| BAT | 24.97 | 8.73 | -\|71.59[*] | -\|35.29[*] | 28.61 | 45.79 | 69.46 | 71.62 | 78.27 | 61.29 |
| **OWL** w/o CoT | 33.31 | 17.24 | 46.15\|77.21[*] | 34.24\|51.67[*] | 24.67 | 31.29 | 74.29 | - | - | 65.27 |
| **OWL** w CoT | **33.37** | **17.26** | **46.17** | **34.31** | **23.29** | **29.91** | **77.89** | **79.04** | **86.76** | **76.53** |

[†] Gemini models are evaluated via API with binaural inputs; results are reported only for event detection. See Appendix D for more details.
[*] As BAT uses a 4-bin protocol, we also report 4-bin results for **OWL** alongside its native 12-bin evaluation (12-bin\|4-bin).

Table 4: **Zero-shot Performance of OWL** on the **SpatialSoundQA** across perception and reasoning tasks. **OWL** consistently outperforms the baselines, with larger gains in spatial reasoning tasks, demonstrating the benefit of the **SAGE** and CoT instruction tuning. Best results are denoted in **bold**.

| Model | Perception (Type ABCD) Detection (mAP)↑ Type A | Type C | DoA (Acc)↑ Type B | Type D | DP (DER)↓ Type B | Type D | Reasoning (Type E) Direction↑ | Distances↑ | Avg↑ |
|---|---|---|---|---|---|---|---|---|---|
| Random | 0.61 | 0.59 | 12.57 | 12.41 | 67.33 | 67.46 | 50.00 | 50.00 | 50.00 |
| Mono BAT | 24.15 | 6.42 | 14.31 | 11.93 | 34.17 | 56.26 | 57.69 | 51.36 | 54.33 |
| BAT | 26.34 | 9.89 | 75.54 | 37.65 | 29.16 | 47.90 | 69.77 | 84.04 | 76.89 |
| **OWL** | **26.76** | **12.73** | **78.31** | **43.15** | **26.14** | **43.21** | **71.21** | **86.91** | **79.06** |

## 6.2 MAIN RESULTS

**SAGE performance on SELD**. Table 2 compares **SAGE** with SELDNet and Spatial-AST on SpatialSoundQA and **BiDepth**. At test time, evaluation uses only binaural audio; depth is incorporated during training as auxiliary supervision. We evaluate three SAGE setups: (i) trained on BiDepth audio, zero-shot on SpatialSoundQA; (ii) pre-trained on **BiDepth** then fine-tuned on SpatialSoundQA; (iii) trained on **BiDepth** with audio & depth, evaluated in-domain and zero-shot. For Spatial-AST, we report models trained on SpatialSoundQA with zero-shot or fine-tuned evaluation on **BiDepth**. SELDNet is trained independently on each dataset and excluded from transfer, as it cannot use depth.

Across both datasets, **SAGE** achieves best or second-best results in nearly every metric. Relative to Spatial-AST, it improves event detection modestly ($\approx 1.6 - 1.7\%$) but yields much larger gains in localization: $ER_{20°}$ decreases by 23.61%, MAE by 25.52%, and DER by 31.34%. In cross-dataset evaluation, SAGE reduces DER by 82% relative to Spatial-AST, while depth supervision mitigates the smaller drops observed in detection and angular metrics. Two main trends emerge: depth supervision consistently strengthens localization, showing that geometry primarily aids spatial reasoning rather than event detection; and **SAGE** transfers far more robustly than baselines, whose performance degrades sharply. Even where mAP gains are modest, **SAGE** remains among the top two models across all metrics, underscoring consistent rather than isolated improvements. Overall, **SAGE** demonstrates clear advantages in localization accuracy and robustness under cross-dataset transfer.

**OWL Performance on QA**. We evaluate **OWL** on **BiDepth** and SpatialSoundQA to assess system-level performance beyond the encoder. On **BiDepth** (Table 3), **OWL** outperforms both closed- and open-source baselines across all task types. Gemini models are included as high-capacity LLM references but only support event detection. Among open-source baselines, BAT is the closest competitor, while RAVEN, VideoLLaMA2, and AudioFlamingo2 provide broader audio-language

comparisons. Leveraging the geometry-aware **SAGE**, **OWL** achieves 46.15% accuracy under fine-grained 12-bin DoA and 77.21% under coarse 4-bin quadrants, compared to BAT's 71.59% (4-bin). Reporting both protocols ensures fairness while highlighting robustness under stricter evaluation. Distance error rates are also lower (24.67% and 31.29%) than the baselines. Reasoning-heavy tasks show the sharpest gains: **OWL** reaches 65.37–74.29% accuracy single-step reasoning and 76.53–77.89% in multi-step CoT reasoning, far surpassing all baselines. Adding CoT supervision further improves reasoning accuracy by 11.26% and yields consistent gains in detection and DoA.

On SpatialSoundQA (Table 4), **OWL** again surpasses BAT, improving DoA accuracy (from 75.54% to 78.31%) and reducing DER (from 29.16 to 26.14). CoT tuning provides the largest boost in reasoning, raising direction and distance accuracy to 71.21% and 86.91% (79.06% overall) compared to BAT's 76.89%. **OWL** is stronger in perception and the first to demonstrate geometry-aware multi-step reasoning. Its encoder provides robust spatial features, while CoT supervision yields explicit, interpretable rationales and consistent benchmark gains. This robust zero-shot result on the unseen dataset ensures no data leakage in **OWL** training. Appendix E presents qualitative examples illustrating CoT efficacy beyond numerical results.

## 6.3    ABLATION STUDY

**Loss Component Ablation for SAGE.** Table 5 quantifies the effect of different loss terms. Training with only the binaural loss $\mathcal{L}_{\text{binaural}}$ ($\eta_2 = 0$) yields mAP = 49.75 but high localization errors (ER$_{20°}$ = 36.89, MAE = 26.32, DER = 17.11). Removing the $\mathcal{L}_{\text{EDC}}$ term ($\lambda = 0$) has little effect on detection but higher impact on localization, indicating that binaural supervision dominates baseline performance.

Table 5: Ablation study of loss components in **SAGE**. Adding the geometric loss $\mathcal{L}_{\text{geo}}$ yields substantial gains in spatial localization (ER$_{20°}$, MAE, DER) while preserving high event detection mAP.

| Loss | | mAP ↑ | ER$_{20°}$ ↓ | MAE ↓ | DER ↓ |
|---|---|---|---|---|---|
| $\mathcal{L}_{\text{binaural}}$ ($\eta_2 = 0$) | | 49.75 | 36.89 | 26.32 | 17.11 |
| $\lambda = 0$ | | 49.73 | 36.79 | 26.12 | 16.71 |
| $\eta_1 = 1e^{-2}$ $\lambda = 1e^{-1}$ | $\eta_2 = 1e^{-4}$ | 49.28 | 36.13 | 25.91 | 21.72 |
| | $\eta_2 = 1e^{-3}$ | 49.39 | 33.64 | 23.31 | 17.47 |
| | $\eta_2 = 1e^{-2}$ | **49.81** | **28.13** | **21.67** | **14.32** |

Small weights on $\mathcal{L}_{\text{geo}}$ marginally reduce ER$_{20°}$ and MAE but destabilize DER. Larger weights ($\eta_1 = 0.01, \eta_2 = 0.001$) improve MAE (23.31) without harming detection. The largest gains occur when $\eta_2 = 0.01$ is applied directly, lowering all errors (ER$_{20°}$ = 28.13, MAE = 21.67, DER = 14.32) while preserving mAP (49.81). Thus, geometry alignment via $\eta_2$ is the main driver of improved localization, while imbalanced or overly small weightings dilute its effect.

**Effect of Training Stage of OWL**. Table 6 evaluates the curriculum stages. Without Stage1 warmup, detection collapses (mAP = 32.92/8.97) and DoA/distance estimation degrades. Adding Stage 1 recovers detection (33.27/17.19) and improves Type II tasks (DoA: 45.91/34.21; Distance: 24.39/31.17), showing that single-source pretraining stabilizes learning. Stage 2 adds relative reasoning, boosting Type III BA to 74.29 and Type IV BA to 65.27, confirming the benefit of explicit geometric supervision. The full three-stage curriculum yields the strongest results: Type III BA = 77.89 and Type IV (Detection 79.04, Direction 86.76, BA 76.53), demonstrating that gradual progression from perception to reasoning is essential for building robust spatial reasoning.

Table 6: Training Stage Ablation of **OWL**. While warmup stage stabilizes training, progressively adding training stages leads to consistent performance improvements across all task types. $^\dagger$ denotes without warmup. Best values are denoted in **bold**.

| Training Stages | | | TypeI | | | | TypeII | | TypeIII | TypeIV | | |
|---|---|---|---|---|---|---|---|---|---|---|---|---|
| | | | Detection (mAP)↑ | | DoA (Acc)↑ | | Distance (DER)↓ | | | | | |
| Stage 1 | Stage 2 | Stage 3 | Single Source | Double source | Single Source | Double source | Single Source | Double source | BA↑ | Detection ↑ | Direction ↑ | BA ↑ |
| ✓$^\dagger$ | ✗ | ✗ | 32.92 | 8.97 | 41.28 | 13.71 | 22.77 | 61.24 | - | - | - | - |
| ✓ | ✗ | ✗ | 33.27 | 17.19 | 45.91 | 34.21 | 24.39 | 31.17 | - | - | - | - |
| ✓ | ✓ | ✗ | 33.31 | 17.24 | 46.15 | 34.24 | 24.67 | 31.29 | 74.29 | - | - | 65.27 |
| ✓ | ✓ | ✓ | **33.37** | **17.26** | **46.17** | **34.31** | **23.29** | **29.91** | **77.89** | **79.04** | **86.76** | **76.53** |

Table 7: **OWL** on real-world binaural acoustic scene classification.

| Class | Street pedestrian | Metro | Park | Street traffic | Metro station | Public square | Airport | Shopping mall | Bus | Tram |
|---|---|---|---|---|---|---|---|---|---|---|
| **Acuracy** | 0.66 | 0.79 | 0.76 | 0.68 | 0.72 | 0.77 | 0.84 | 0.71 | 0.83 | 0.77 |

## 6.4 REAL-WORLD GENERALIZABILITY

To evaluate whether **OWL** relies on synthetic data and to examine its behavior in realistic acoustic conditions, we conducted additional experiments using the real-world DCASE Binaural Audio Scene Classification dataset Mars et al. (2019) and the DCASE SELD Challenge 2021 dataset Politis et al. (2021). These datasets were selected because both are recorded in real environments. The first supports a general audio perception task, and the second supports a spatial localization task. Neither dataset is used during training, and both contain diverse environments that differ from our synthetic domains in microphone characteristics, background noise patterns, and reverberant structure.

**Audio Scene Classification**. For zero-shot evaluation on the audio scene classification dataset, we used the instruction-based prompt without any adaptation or fine-tuning. The model achieves stable accuracy across all ten acoustic scenes (Table 7), including street pedestrian 0.66, metro 0.79, park 0.76, street traffic 0.68, metro station 0.72, public square 0.77, airport 0.84, shopping mall 0.71, bus 0.83, and tram 0.77. These results show that **OWL** generalizes well to real-world binaural recordings, despite being trained primarily on synthetic spatial audio. This behavior indicates that the geometry-aware audio encoder and the downstream reasoning components capture structure that transfers across simulation-to-real domain gaps.

Table 8: **OWL** on real-world acoustic source localization task.

| Method | Type I | | | |
|---|---|---|---|---|
| | Detection (mAP) (↑) | | DoA (Acc) (↑) | |
| | Single source | Double source | Single source | Double source |
| **OWL** | 57.21 | 51.26 | 42.78 | 31.46 |
| **OWL** + CoT | 57.19 | 51.09 | 42.63 | 31.29 |

**Sound Source Localization**. To further demonstrate the compatibility of **OWL** with spatial reasoning tasks on real-world audio, we evaluate it on the DCASE SELD Challenge 2021 dataset. Because this dataset is recorded in first-order ambisonic format, we convert the FOA signals to binaural audio using an HRTF transformation Noisternig et al. (2012). This step is required because the **SAGE** encoder operates on binaural input and no real-world binaural spatial dataset is publicly available. Table 8 reports the performance of **OWL** on this dataset. **OWL** achieves consistent performance across both single-source and double-source conditions, which shows **OWL**'s effectiveness in these real-world settings.

## 7 CONCLUSION & FUTURE WORKS

We introduced **BiDepth**, a large-scale dataset, and **OWL**, the first spatial audio LLM with geometry-aware Chain-of-Thought reasoning. By coupling binaural audio with panoramic depth supervision, **OWL** achieves strong gains in event detection, DoA, distance estimation, and higher-order spatial reasoning. Ablations show that geometry chiefly strengthens localization, while CoT alignment is essential for multi-step reasoning, together yielding interpretable and transferable representations.

A key limitation is that **BiDepth** is simulation-based, leaving open the question of robustness in real-world acoustic conditions. Moreover, our reasoning tasks are restricted to single-turn QA, whereas human spatial communication is interactive and dialog-based. Future work will extend **BiDepth** to real recordings, expand reasoning to multi-turn dialogues, and explore richer grounding with vision or inertial sensing. Preference-based alignment could also improve the coherence of generated rationales. Taken together, these directions position **OWL** as a step toward embodied agents capable of human-like spatial reasoning.

ACKNOWLEDGMENT

We acknowledge support from the National Institute on Drug Abuse (1R01DA059422) and the National Institute of Diabetes and Digestive Kidney Diseases (5R01DK138866). Results in this paper were obtained in part using a high-performance computing system acquired through NSF MRI grant DMS-1337943 to WPI. We gratefully acknowledge their support in enabling this work.

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

APPENDIX

# A  BIDEPTH CURATION DETAILS

## A.1  DATASET STATISTICS

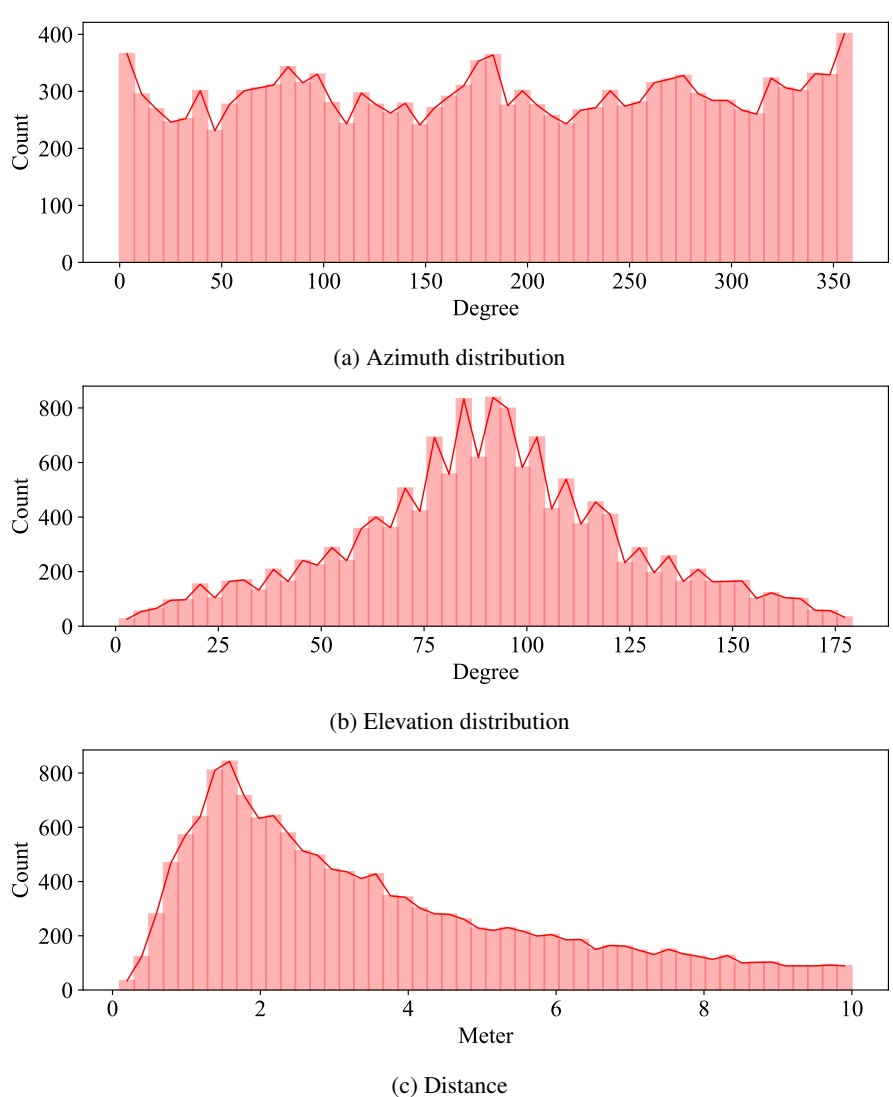

(a) Azimuth distribution

(b) Elevation distribution

(c) Distance

Figure 5: Distributions of azimuth, elevation, and source-receiver distance in **BiDepth**. Azimuth angles are nearly uniform, elevation is skewed toward the horizontal plane (reflecting typical indoor acoustics), and distances peak around 1.8 m within a 10 m range. The dataset, comprising 28K binaural RIRs and 1.1M QA pairs, will be made publicly available to ensure reproducibility and facilitate further research.

Statistical analysis of spatial distributions validates that **BiDepth** provides balanced and diverse coverage for training and evaluation of geometry-aware models. Figure 5 reports the distributions of azimuth, elevation, and distance for sound sources relative to the receiver. Azimuth angles are nearly uniform across $[0°, 360°)$, confirming that the dataset covers the full horizontal plane without bias toward particular directions. Elevation angles follow a unimodal distribution centered near $92°$, corresponding to sources located close to the horizontal plane of the receiver. While this distribution is skewed toward horizontal orientations, it mirrors realistic indoor acoustics where most sound sources (e.g., speakers, televisions, human speech) occur around ear level. Source-receiver distances

are bounded within 10 meters, with the highest density at approximately 1.8 meters. This ensures the majority of pairs are in-room, while still including out-of-room placements that require models to reason about occlusion and indirect propagation.

Overall, **BiDepth** consists of 28K unique binaural RIRs and panoramic depth maps, paired with over 1.1M QA examples. The distribution of QA types (detection, direction estimation, spatial reasoning, and CoT reasoning) is reported in Figure 6, confirming that each reasoning level is well represented. Although synthetic, the scale of **BiDepth** exceeds prior spatial audio corpora by an order of magnitude, providing the coverage needed for geometry-aware training and evaluation.

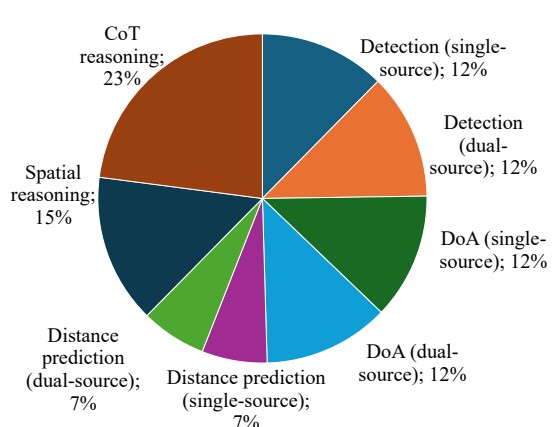

Elevation distribution is biased toward horizontal sources, potentially underrepresenting extreme overhead or below-floor conditions. Nonetheless, this bias reflects realistic indoor settings and supports stable training. Extending **BiDepth** to environments with more diverse vertical layouts (e.g., multi-floor or outdoor spaces) is a promising direction for future work. The diversity captured here, including in-room and out-of-room scenes, underpins the generalization results reported in Section 6.

Figure 6: Distribution of question types in **BiDepth**, including detection (single/dual-source), direction-of-arrival (DoA), distance prediction, spatial reasoning, and chain-of-thought (CoT) reasoning, shows balanced representation of each category.

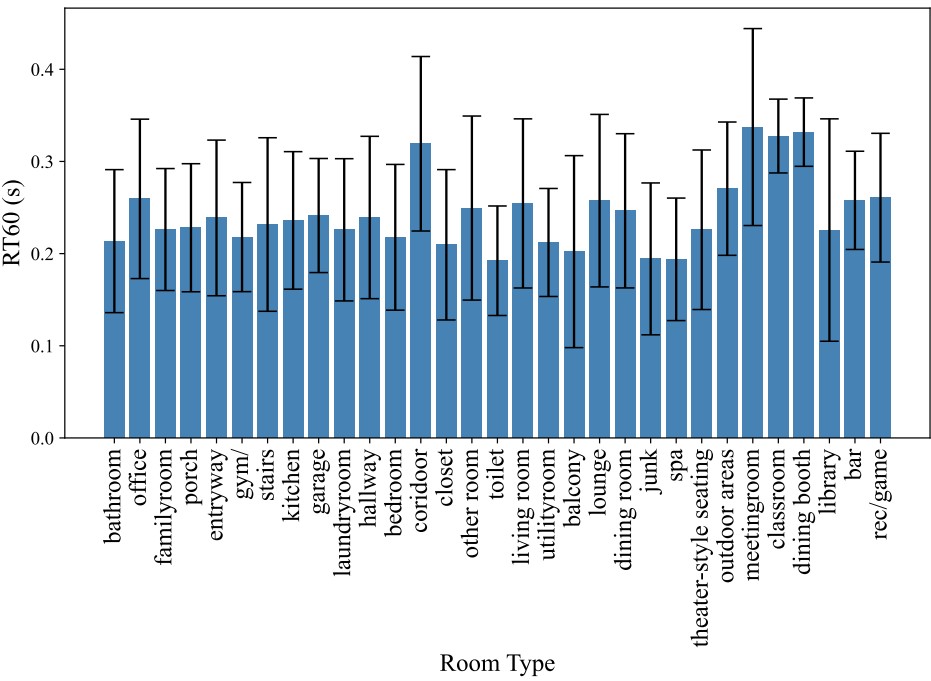

Figure 7: **RT60 distribution across different room types**, showing the variation in reverberation times with error bars indicating standard deviation. Higher RT60 values correspond to more reverberant environments, while lower values indicate faster sound decay.

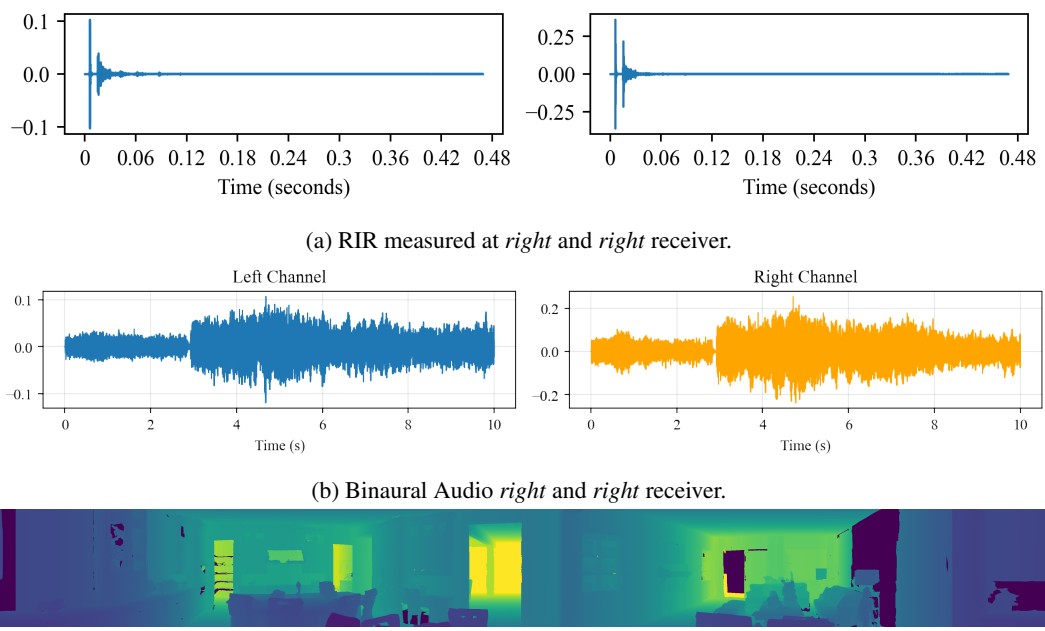

(a) RIR measured at *right* and *right* receiver.

(b) Binaural Audio *right* and *right* receiver.

(c) Panoramic depth image of environments at $20^o$ FOV, captured from receiver position $r$.

Figure 8: Example binaural RIR-depth image from **BiDepth**. (a) Binaural room impulse responses (RIRs) at the left and right receivers. (b) Binaural Audio at left and right receiver (c) Panoramic depth image constructed from a $20°$ field-of-view sweep at the receiver position.

## A.2 RT60 Statistics of Different Rooms

Statistical analysis of reverberation confirms that **BiDepth** captures realistic acoustic diversity across room types. Figure 7 reports RT60 values across different Matterport3D Chang et al. (2017) environments used for simulation.

RT60 is a standard acoustic measure defined as the time required for the sound pressure level to decay by 60 dB Hak et al. (2012). Values in **BiDepth** range from approximately 0.1-0.4s, consistent with typical furnished indoor spaces such as bedrooms, offices, and kitchens, while remaining shorter than highly reverberant environments like concert halls or auditoriums. Variation arises from differences in room size, object arrangement, and construction materials. Importantly, the dataset includes both in-room and out-of-room configurations, where indirect propagation paths yield longer decay times, further enriching reverberation diversity.

A limitation is that RT60 values are derived from simulated Matterport3D settings, which may underrepresent extreme real-world cases with very high reverberation. Extending the dataset to cover more reverberant environments, particularly measured in real-world settings, is an important direction for future work.

## A.3 Example of binaural RIR and Depth Image

Figure 8 presents a representative example of the paired acoustic and geometric signals generated in our simulation pipeline. Binaural RIRs capture spatialized cues at the receiver's left and right ears, while the panoramic depth image encodes the surrounding geometry. These paired signals are central to **SAGE**, where depth-guided supervision regularizes the audio representation and enables geometry-aware learning. While this example is simulated, it reflects realistic variations in propagation and geometry diversity observed in indoor environments.

### A.4 QUESTION-ANSWER DATASET CURATION DETAILS

We generate four types of question-answer pairs that consist of event detection, direction estimation, spatial reasoning, and reasoning with CoT.

**Type I**. We curate **135K** QA pairs for the single-source setting, where each question requires the model to identify the event category. A few representative question formats used in this setting are illustrated below.

---

**Type I Questions (Single Source)**

- Can you tell me what kind of sounds are in this recording?
- What categories of sounds are present here?
- Could you list the types of sounds captured in this audio?
- Which sound events are audible in this clip?
- Please identify all the distinct sounds in this recording.
- Can you break down the audio into individual sound events?

---

We then curate another **135K** QA pairs with two sound sources. In this setting, given the location of one source, **OWL** is required to predict its event category. Representative question formats are shown below, where each query follows the {o'clock}, {up/down}, {distance}m convention.

---

**Type I Questions (Dual Source)**

- Can you categorize the sounds in the audio that are located to the {}, {}, at an estimated distance of {} meters?
- Determine the types of sounds present in the audio clip from directions to the {}, {}, approximately {} meters distant.
- Enumerate the sound occurrences in the audio clip that are sourced from the {}, {}, around {} meters away.
- Point out the sound sources heard from {}, {} at an approximate distance of {} meters,
- Can you pick out the sound events originating from {}, {} approximately {} meters away?.
- What sound sources can be identified from {}, {} roughly {} meters distant?

---

**Type II**. We create **135K** single-source QA pairs for estimating the direction of arrival given a sound source class. The answers follow the format `o'clock; up/down; distance m`. Representative question types are illustrated below.

---

**Type II Questions (Single Source)**

- Could you identify the likely area or setting of this sound clip's source?
- What is the most likely place or scene producing this sound clip?
- What's the spatial origin of this sound clip?
- Which sound events are audible in this clip?
- Where do you think this audio clip originates?
- Could you pinpoint the approximate region or context of the sound clip's source?

---

We further curate **135K** two-source QA pairs for localization. Similar to Type I, the class of one sound event is provided, and **OWL** is required to estimate its location. Representative examples of this question type are shown below, where `<CLS>` is replaced with the corresponding sound event class.

---

**Type II Questions (Dual Source)**

- At what distance and in which direction, is the `<CLS>` sound originating?
- At what spot is the sound of the `<CLS>` audible?
- Where, in terms of direction and distance, can the sound of the `<CLS>` be located?
- Can you estimate the bearing and distance of the `<CLS>`'s sound source?
- How would you locate the `<CLS>`'s sound in terms of both distance and direction from you?
- What is the direction and range of the `<CLS>`'s sound origin?

---

**Type III**. We curate **300K** QA pairs targeting relational reasoning, including leftright, frontback, abovebelow, and distance comparisons between two sources. These questions emphasize richer geometric understanding beyond basic perception. All corresponding binaural audio samples contain two sound sources. The answers to these questions are binaray (Yes/No). Representative question formats are illustrated below. We replace the {} with sound event class names.

---

**Type III Questions**

- Are the sounds of {} coming from the left of you?
- Is the sound of {} coming from overhead?
- Are the sounds of {} coming from your back?
- Are the sounds of {} coming from the left of the sound of {}?
- Is the sound of {} located in front compared to ?
- Are {} and {} both originating from one side of you?

---

**Type IV**. We curate **250K** Chain-of-Thought (CoT) QA pairs, where each answer includes stepwise reasoning in addition to the final prediction. These questions extend beyond perception and relational queries by requiring explicit multi-step inference grounded in both acoustic and geometric cues. As with Type III, all binaural audio samples contain two sound sources. Representative CoT question–answer pairs format are provided below.

---

**Type IV QA Pairs (left-right)**

**Question:** Which sound can be heard to the left of the receiver?
**Answer format (One left):** `<s1>` originates from `<s1p>` while `<s2>` is at `<s2p>`. Therefore, `<s1>` is on the left side of the receiver.
**Answer format (One right):** Relative to the receiver, `<s1>` comes from `<s1p>` and `<s2>` from `<s2p>`. This shows that `<s2>` lies on the right.
**Answer format (Both left):** Both `<s1>` and `<s2>` are positioned at `<s1p>` and `<s2p>`, which lie to the left of the receiver.
**Answer format (Both right):** Since `<s1>` is at `<s1p>` and `<s2>` is at `<s2p>`, and both positions are on the right, the two sounds lie on the right side of the receiver.

---

**Type IV QA Pairs (front-back)**

**Question:** From the receiver's perspective, which sound originates ahead?
**Answer format (One front):** `<s1>` originates from `<s1p>` while `<s2>` is at `<s2p>`. Therefore, `<s1>` is on the left side of the receiver.
**Answer format (One back):** `<s1>` is located at `<s1p>`, while `<s2>` is positioned at `<s2p>`. Therefore, `<s2>` is behind the receiver.
**Answer format (Both front):** Since `<s1>` comes from `<s1p>` and `<s2>` from `<s2p>`, both sounds are in front of the receiver.
**Answer format (Both back):** Because `<s1>` originates from `<s1p>` and `<s2>` from `<s2p>` both sources are located at the back side.

> **Type IV QA Pairs (up-down)**
>
> **Question:** Identify the sound that is located on the upper side of the receiver.
> **Answer format (One up):** `<s1>` is at `<s1p>`, while `<s2>` is at `<s2p>`. Therefore, `<s1>` is coming from above the receiver.
> **Answer format (One down):** With respect to the receiver, `<s1>` at `<s1p>` and `<s2>` at `<s2p>` indicate that `<s2>`is on the lower side.,
> **Answer format (Both up):** Because `<s1>` originates from `<s1p>` and `<s2>` from `<s2p>`, both are situated on the upper side.
> **Answer format (Both down):** Since `<s1>` is at `<s1p>` and `<s2>` at `<s2p>`, both sounds are located beneath.

We replace `<s1>`, `<s2>`, `<s1p>`, and `<s2p>` with source 1 label, source 1 position, source 2 label, and source 2 position.

## A.5 EXAMPLE OF QUESTION-ANSWER PAIR

Table 9: Examples of QA pairs from **BiDepth** across the four task types (I-IV). The samples span event detection, localization, relative spatial reasoning, and Chain-of-Thought (CoT) supervision.

| Type | No. of Sound Source | Objective | Question-Answer |
|---|---|---|---|
| I | 1 | Event Detection | **Q:** Could you describe the various sounds detected here? 
 **A:** Male singing; Child singing |
| | 2 | Given a Direction, detect the sound event | **Q:** Identify the sound events in the audio clip coming from the one o'clock, down, approximately 3.0, meters away. 
 **A:** Bicycle; Bicycle bell |
| II | 2 | Direction Estimation | **Q:** From which direction and at what distance can the sound of the Speech be detected? 
 **A:** seven o'clock; down; 0.5 |
| | | Given a sound source, estimate the location | **Q:** Which way and how far off is the Bird flight, flapping wings sound's origin? 
 **A:** six o'clock; down; 1.5 m |
| III | 2 | Left-right composition | **Q:** Are the sounds of Speech coming from the left of you? 
 **A:** Yes |
| | | Front-back composition | **Q:** Is the Speech sound originating from the rear? 
 **A:** No |
| | | Upper-below composition | **Q:** Is the sound of Speech originating from below in relation to Speech? 
 **A:** Yes |
| | | Comparing distance between two sound sources | **Q:** Does the Music sound arrive from a smaller direct distance than the Rodents, rats, mice sound? 
 **A:** Yes |
| | | | **Q:** Is Music coming from a nearer point than Music when considering a direct path? 
 **A:** No |
| | | | **Q:** Is the origin of Boat, Water vehicle sound located closer than the origin of Speech sound? 
 **A:** No |
| IV | 2 | Left-right composition | **Q:** What sound originates from the receiver's left side? 
 **A:** Relative to the receiver, Wheeze and Sawing are detected at eight o' clock and nine o' clock. Thus, they both lie on the left. |
| | | Front-back composition | **Q:** Identify the sound that is located on the lower side of the receiver. 
 **A:** Because Pulleys comes from nine o' clock and Animal from two o' clock, the source below is Animal. |
| | | Upper-below composition | **Q:** Which sound can be heard in front of the receiver? 
 **A:** Relative to the receiver, Speech and Speech at nine o' clock and four o' clock do not correspond to front directions. |

Table 9 shows representative QA examples from **BiDepth** across the four task types. These span event detection (TypeI), localization with direction and distance (TypeII), relational reasoning (TypeIII), and explicit step-by-step rationales (TypeIV, Chain-of-Thought). Together, they illustrate the progression from low-level perception to geometry-aware multi-step reasoning. The taxonomy aligns with the staged training curriculum of **OWL**, ensuring that each reasoning level is explicitly supervised. As

detailed in the main paper, the four types are represented in roughly equal scale, allowing controlled evaluation across perception and reasoning.

# B MODEL DETAILS

## B.1 FEATURE EXTRACTION FROM BINAURAL AUDIO

We follow prior work Zheng et al. (2024b) and extract two complementary representations from the binaural waveform $B^r(t)$. A binaural recording is represented as

$$B^r(t) = \begin{bmatrix} B^r_L(t) \\ B^r_R(t) \end{bmatrix} = \begin{bmatrix} \text{RIR}_L(t, s, r, \gamma) \\ \text{RIR}_R(t, s, r, \gamma) \end{bmatrix} \circledast M^s(t),$$

where $B^r_L(t)$ and $B^r_R(t)$ are the left and right ear signals, $M^s(t)$ is the monaural input event, $\text{RIR}_n(t, s, r, \gamma)$ is the room impulse response from source $s$ to receiver $r$ under configuration $\gamma$, and $\circledast$ denotes convolution.

We first compute the short-time Fourier transform (STFT) for each channel $c \in \{L, R\}$:

$$X_c(m, k) = \sum_{n=0}^{N-1} B^r_c[n]\, w[n - mH]\, e^{-j2\pi kn/N},$$

where $w[\cdot]$ is a window of length $N = 1024$, $H = 320$ is the hop size, $m$ indexes time frames, and $k$ indexes frequency bins.

From the STFT magnitudes, we derive the **Mel-spectrogram**. With filterbank $\mathbf{M} \in \mathbb{R}^{F \times K}$ (with $F = 128$ Mel bands and $K = 512$ frequency bins), the Mel energy at frame $m$ and band $f$ is

$$S_c(m, f) = \sum_{k=1}^{K} \log\big(M(f, k)\, |X_c(m, k)|^2\big).$$

To encode spatial cues, we compute the Interaural phase difference (IPD):

$$\text{IPD}(m, k) = \angle \frac{X_L(m, k)}{X_R(m, k)}.$$

To avoid phase wraparound instabilities, IPD is represented using sine and cosine transforms:

$$\text{IPD}_{\cos}(m, k) = \cos(\text{IPD}(m, k)), \quad \text{IPD}_{\sin}(m, k) = \sin(\text{IPD}(m, k)).$$

These are filtered with the same Mel filterbank:

$$\widetilde{\text{IPD}}_{\cos}(m, f) = \sum_{k=1}^{K} M(f, k)\, \text{IPD}_{\cos}(m, k), \quad \widetilde{\text{IPD}}_{\sin}(m, f) = \sum_{k=1}^{K} M(f, k)\, \text{IPD}_{\sin}(m, k).$$

Finally, we concatenate the four feature maps into the input tensor:

$$\mathcal{Z} = [S_L, S_R, \widetilde{\text{IPD}}_{\cos}, \widetilde{\text{IPD}}_{\sin}], \quad \mathcal{Z} \in \mathbb{R}^{4 \times M \times F},$$

where $M$ is the number of time frames, $F$ is the number of Mel bands, and the leading dimension $4$ corresponds to the left Mel, right Mel, and IPD sine/cosine channels. This representation preserves both semantic information (via Mel energy) and geometric cues (via phase differences), providing a strong input basis for **SAGE**.

## B.2 SAGE ARCHITECTURE DETAILS

**SAGE** consists of two modules: a binaural audio encoder and an RIR prediction module, trained jointly with geometric supervision.

The **binaural audio encoder** processes $\mathcal{Z}$ through an initial $3 \times 3$ 2D convolution, batch normalization, and GELU. Features are then patch-embedded using a $16 \times 16$ CNN in both time and frequency,

producing non-overlapping tokens. These are passed through a 12-layer Transformer encoder (hidden size 768, 12 heads). Three separate linear heads, each attached to [CLS] tokens, predict event class, direction of arrival, and distance.

The **RIR prediction module** processes panoramic depth images using a ResNet-18 encoder. The resulting depth features are fused with the audio encoder's output and passed to a ResNet-18 decoder with transposed convolution to reconstruct the binaural RIR. This auxiliary task supplies privileged geometric supervision, improving spatial reasoning of the binaural audio encoder of **SAGE** without requiring depth at inference.

### B.3 **OWL** ARCHITECTURE DETAILS

We reuse the **SAGE**'s binaural audio encoder as our binaural front end. The network is a 12-layer Transformer encoder with **85.52M** parameters. Given a binaural input, the encoder produces a sequence of frame-level embeddings that capture spectral, interaural, and temporal cues needed for downstream spatial inference. To interface the acoustic features with the language model, we adopt a Q-Former as a learned projector. The module has **8** Q-Former layers and a bank of **64** learnable queries. It is trained *from scratch* to attend over the encoder outputs and emit a compact set of query–aligned tokens. we use **LLaMA-2-7B** as the backbone. We fine-tune it with **LoRA** on the *Query* and *Value* projection matrices inside the self–attention blocks, using rank $r=8$ and $\alpha=32$. This adaptation introduces **4.1M** trainable parameters, which is roughly **0.062%** of the total model size, while the remaining weights stay frozen. The resulting tuning strategy preserves the linguistic competence of LLaMA-2-7B and focuses capacity on aligning the model to **OWL**'s audio-geometric tokens.

## C TRAINING DETAILS

### C.1 TRAINING PROCEDURE OF **SAGE**

Table 10: Hyperparameters used to train **SAGE**, including dataset source, optimizer settings, learning rate schedule, and training hardware configuration.

| Description | Value |
|---|---|
| Sound Source | AudioSet-2M |
| Audio Normalization | Loudness |
| Augmentation | Yes |
| Weighted Sampling | Yes |
| Optimizer | AdamW Loshchilov & Hutter (2016) |
| Optimizer Momentum | $\beta_1 = 0.9$, $\beta_2 = 0.95$ |
| Weight Decay | 0.0001 |
| Base learning rate | 0.001 |
| Learning rate schedule | Half-cycle Cosine Loshchilov & Hutter (2017) |
| GPU | $4 \times$ A100 |

We train **SAGE** in two sequential stages to progressively incorporate both classification and geometric supervision.

**Stage 1 (Audio Pretraining)**. This stage focuses solely on the binaural audio encoder to ensures stable audio representations before introducing additional supervision. The binaural encoder is initialized from AudioMAE Huang et al. (2022) and fine-tuned for 40 epochs using only the event classification loss $\mathcal{L}_{\text{cls}}$. This loss encourages the encoder to learn discriminative features for sound event recognition, serving as a stable initialization point for subsequent joint optimization.

**Stage 2 (Joint Training)**. We attach the RIR prediction module to binaural audio encoder and optimize both components together for 60 epochs, enabling the network to strike a balance between perceptual classification and geometry-aware learning. The training objective is

$$\mathcal{L} = \eta_1 \mathcal{L}_{\text{binaural}} + \eta_2 \mathcal{L}_{\text{geo}},$$

where $\mathcal{L}_{\text{binaural}} = \alpha_1 \mathcal{L}_{\text{cls}} + \alpha_2 \mathcal{L}_{\text{dis}} + \alpha_3 \mathcal{L}_{\text{doa}}$. We set $\eta_1 = 1$, $\eta_2 = 0.01$, and $\alpha_1 = 1$ for audio pretraining stage and $\alpha_1 = 1250$ in joint training, $\alpha_2 = 1$, and $\alpha_3 = 2$ unless otherwise noted in

Table 11: Hyperparameters used for training **OWL**, including optimization settings, LoRA configuration, and stage-wise epoch schedule.

| Description | Value |
|---|---|
| Sound Source | AudioSet-20K |
| Audio Normalization | Loudness |
| Augmentation | No |
| Weighted Sampling | No |
| LLM backbone | LLaMA-2-7B Touvron et al. (2023) |
| Optimizer | AdamW Loshchilov & Hutter (2016) |
| Epochs | Stage-1: 2
Stage-2: 2
Stage-3: 3 |
| Learning Rate | 0.0001 |
| LoRA Rank | 8 |
| LoRA alpha | 32 |
| Global batch size | 8 |
| GPU | $4 \times$ A100 (80 GB) |

ablations. The hyperparameters used for both stages of training are summarized in Table 10. To understand the effect of different hyperparameter configurations, we perform ablation studies, which are reported in section 6.3. This design ensures that the encoder not only captures task-relevant event information but also aligns its latent space with geometric cues reflected in room impulse responses.

**Optimization**. We use AdamW with an initial learning rate of $1 \times 10^{-4}$, cosine decay, weight decay of $0.01$, and gradient clipping at $1.0$. A linear warm-up is applied for the first 5k steps of Stage 1. Batch size is 64. Training is performed on $4 \times$A100 GPUs with mixed precision (fp16).

## C.2   TRAINING PROCEDURE OF **OWL**

We adopt the three-stage curriculum described in Section 5.1, progressively transitioning from perception to relative reasoning to Chain-of-Thought supervision. The goals are to stabilize optimization early, introduce progressively harder supervision, and restrict learning to adapters while preserving the pretrained encoders.

**Stage 1**. The model is trained for two epochs on Type I-II QA pairs, with 5000 steps of half-cycle cosine LR warm-up.

**Stage 2**. The model continues to train for 2 epochs on Type III reasoning pairs.

**Stage 3**. The model is trained for three epochs on Type IV QA pairs with CoT rationales.

**Frozen vs trainable modules**. The binaural encoder $\phi_a(\cdot)$ is frozen throughout the stages. The projection module $\psi(\cdot)$ is trained from scratch. The LLaMA-2-7B decoder is adapted with LoRA Hu et al. (2022) applied to query, key, and value projections in all attention layers. LoRA rank is 8, scaling factor $\alpha = 32$, and dropout 0.05, yielding $\sim 0.8\%$ trainable parameters. This ensures efficient adaptation with limited parameters. This strategy gradually stabilizes, refines, and consolidates the reasoning ability of **OWL**. Table 11 summarizes the hyperparameters used in training.

**Optimization**. Training uses AdamW with initial learning rate $1 \times 10^{-4}$, cosine decay, weight decay 0.01, and gradient clipping 1.0. Batch size is 32 due to LLM memory limits. Training runs on $4 \times$A100 GPUs with mixed precision.

## D  GEMINI API RESPONSE EXAMPLE

```python
import os
from google import genai

class Gemini:
    def __init__(self, model):
        self.api_key = os.environ["GEMINI_API_KEY"]
        self.client = genai.Client(
            api_key=self.api_key
        )
        self.audio_dir = audio_dir
        self.model = model

    def get_response(self, binaural_audio_file_path, question):
        file = self.client.files.upload(
            file=binaural_audio_file_path
            )
        response = self.client.models.generate_content(
            model=self.model,
            contents=[question, file]
        )
        print(response.text)

if __name__ == "__main__":
    parser = argparse.ArgumentParser()
    parser.add_argument("--api_key", type=str, required=True)
    parser.add_argument("--model", type=str, required=True)
    args = parser.parse_args()

    os.environ["GEMINI_API_KEY"] = args.api_key
    cls = Gemini(model=args.model)

    file_path = "path/to/binaural_audio_file.mp3"
    question = "Where does the Car's sound originate in relation to your
                                        position?"
    answer = cls.get_response(
        binaural_audio_file_path=file_path,
        question=question
    )
```

For an audio sample where the ground-truth car `sound location` is `9 o'clock; 5.0 m`, we obtain the following model response:

```
The car's sound originates from a source that is approaching you
rapidly passing by and then moving away into the distance.
The strong **Doppler effect** is very clear: the pitch and volume
rise sharply as it approaches, reach a peak as it passes
your position, and then quickly decrease as it moves away.
```

# E    QUALITATIVE RESULTS

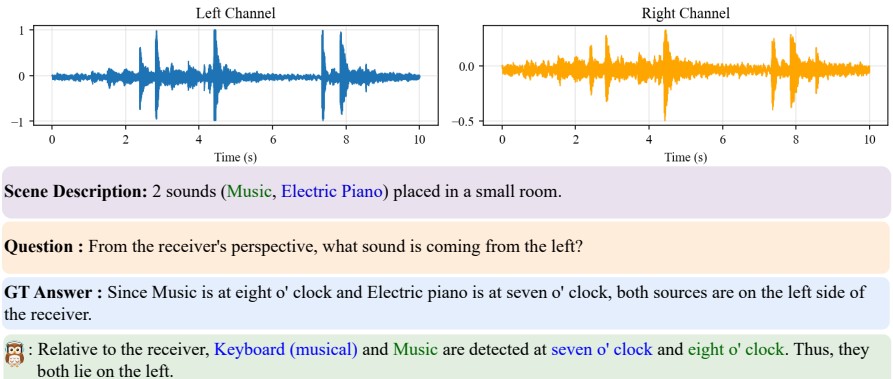

**Scene Description:** 2 sounds (Music, Electric Piano) placed in a small room.

**Question :** From the receiver's perspective, what sound is coming from the left?

**GT Answer :** Since Music is at eight o' clock and Electric piano is at seven o' clock, both sources are on the left side of the receiver.

: Relative to the receiver, Keyboard (musical) and Music are detected at seven o' clock and eight o' clock. Thus, they both lie on the left.

Figure 9: **Example of a left-right spatial reasoning question.** Two concurrent sounds (Music and Electric Piano) are placed in a small room, and the system identifies both as originating from the left side of the receiver.

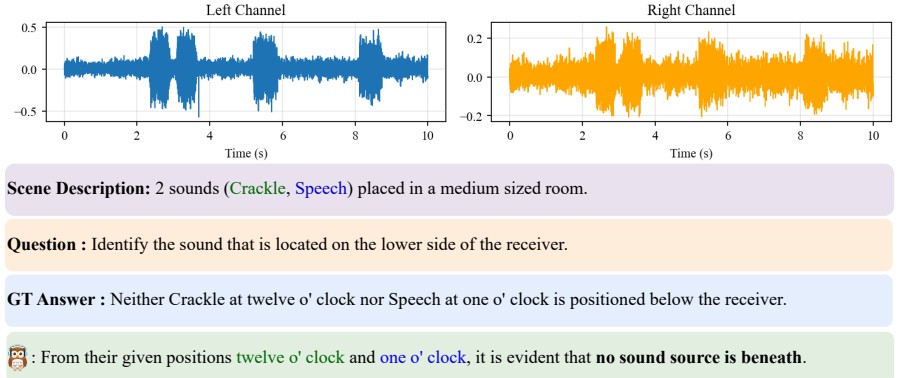

**Scene Description:** 2 sounds (Crackle, Speech) placed in a medium sized room.

**Question :** Identify the sound that is located on the lower side of the receiver.

**GT Answer :** Neither Crackle at twelve o' clock nor Speech at one o' clock is positioned below the receiver.

: From their given positions twelve o' clock and one o' clock, it is evident that **no sound source is beneath**.

Figure 10: **Example of an up-down spatial reasoning question.** Two sounds (Crackle and Speech) are positioned at twelve o'clock and one o'clock, leading to the conclusion that no source is located beneath the receiver.

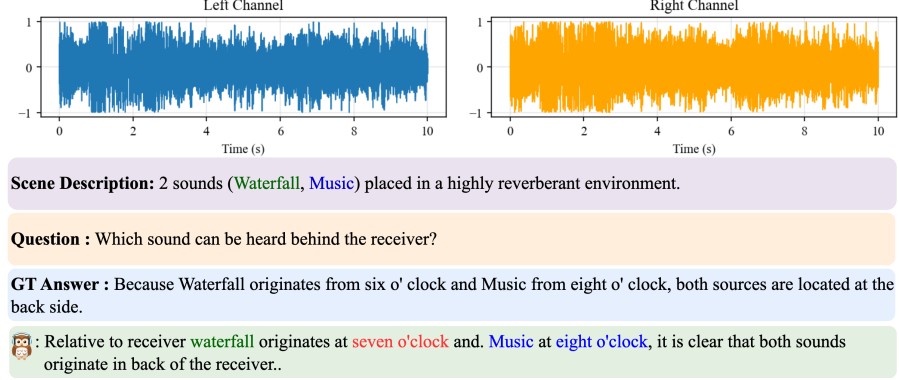

**Scene Description:** 2 sounds (Waterfall, Music) placed in a highly reverberant environment.

**Question :** Which sound can be heard behind the receiver?

**GT Answer :** Because Waterfall originates from six o' clock and Music from eight o' clock, both sources are located at the back side.

: Relative to receiver waterfall originates at seven o'clock and. Music at eight o'clock, it is clear that both sounds originate in back of the receiver..

Figure 11: **Example of a back-front spatial reasoning question.** Two sounds (Waterfall and Music) are placed in a highly reverberant environment. While reverberation causes a slight error in localizing the Waterfall (seven instead of six), the final reasoning still correctly infers that both sources are behind the receiver.

## F    EVALUATION METRIC DETAILS

**Mean Average Precision (mAP)**. Mean Average Precision evaluates ranking quality by averaging precision across all relevant retrievals. For each query, the average precision (AP) is computed, and the overall mAP is given by

$$\text{mAP} = \frac{1}{N} \sum_{i=1}^{N} \text{AP}_i, \tag{5}$$

where $N$ is the number of queries. Higher values indicate better consistency in retrieval or classification.

**Direction of Arrival (DoA) Estimation.**. We report two metrics for spatial localization:

*Mean Angular Error (MAE)* This measures the average angular distance between predicted $(\hat{\theta}, \hat{\phi})$ and ground-truth $(\theta, \phi)$ directions:

$$\text{MAE} = \frac{1}{N} \sum_{i=1}^{N} \Delta \alpha_i, \tag{6}$$

where $\Delta \alpha_i$ is computed using the spherical law of cosines.

*Error Rate at $20°$ (ER$_{20°}$).* This is the fraction of samples whose angular error exceeds $20°$:

$$\text{ER}_{20°} = \frac{1}{N} \sum_{i=1}^{N} \mathbb{1}\left[\Delta \alpha_i > 20°\right]. \tag{7}$$

**Distance Estimation**. We evaluate distance prediction using the Distance Error Rate (DER), defined as the proportion of samples where the predicted distance deviates from ground truth by more than 0.5 m:

$$\text{DER} = \frac{1}{N} \sum_{i=1}^{N} \mathbb{1}\left[|\hat{d}_i - d_i| > 0.5\right], \tag{8}$$

where $\hat{d}_i$ and $d_i$ denote the predicted and ground-truth distances, respectively.

**Binary Accuracy (BA)**. Binary accuracy measures the proportion of samples where the predicted binary label matches the ground truth. Formally, given predictions $\hat{y}_i \in \{0, 1\}$ and ground truth labels $y_i \in \{0, 1\}$, it is defined as

$$\text{BA} = \frac{1}{N} \sum_{i=1}^{N} \mathbb{1}\left[\hat{y}_i = y_i\right], \tag{9}$$

where $N$ is the total number of samples, and $\mathbb{1}[\cdot]$ denotes the indicator function.

## G    USE OF LLM

We used the **Google Nono-Banana** model to generate the logo and Figure 1 (acoustic environment), and the **ChatGPT 5** model to generate Figure 2. All other LLM usage is explicitly documented in the paper where relevant.

