# OpenReview forum: "OWL : Geometry-Aware Spatial Reasoning for Audio Large Language Models"
_ICLR.cc/2026/Conference — ICLR 2026 Poster_

### Official Review · Reviewer_V2cc · 2025-10-31

**Soundness:** 3
**Presentation:** 3
**Contribution:** 3
**Rating:** 6
**Confidence:** 3

**Summary:**

This paper introduces a novel framework for spatial audio understanding, marked by three core contributions. First, it presents BiDepth, a new large-scale synthetic dataset for spatial audio tasks. Second, it proposes SAGE, a novel spatial audio encoder designed to align binaural acoustic features with 3D spatial structure. Finally, it introduces OWL, an audio LLM that utilizes the SAGE encoder to extract and process acoustic and spatial features. The integrated OWL model is shown to achieve superior performance on both the proposed BiDepth dataset and the existing SpatialSound-QA benchmark.

**Strengths:**

- The development of the BiDepth dataset is a significant contribution to the field. A large-scale, well-structured dataset for spatial audio understanding can facilitate further research.
- The SAGE encoder is also a good design.

**Weaknesses:**

- Lack of Evaluation on Real-World Data. The model's performance is exclusively validated on simulated data. While the results are encouraging, models trained solely on synthetic data often fail to generalize to real-world scenarios due to the domain gap. The paper would be significantly strengthened by testing the model's robustness and transferability on real-world datasets, such as those from the DCASE Challenge on spatial audio tasks.

**Questions:**

- What is the model's performance on real-world data? For example, on the DCASE Challenge?
- In the results presented in Table 2, the model using Chain-of-Thought (CoT) reasoning shows improved performance on Type I and Type II tasks. This is counterintuitive, as these tasks do not seem to require complex reasoning. Why CoT is beneficial for these tasks?
- What is the performance of the SAGE encoder for general audio perception tasks? Besides, what are the advantages of a unified encoder like SAGE, which models audio and spatial information jointly, compared to a multi-encoder architecture that models them separately? For instance, [1] demonstrates success with separate modeling. Have you experimented with combining a state-of-the-art audio encoder with the SAGE encoder?

[1] Tang, C.,et al. Can Large Language Models Understand Spatial Audio? Proc. Interspeech 2024

---

> ### Author Response · Authors · 2025-11-19
> **Response to reviewer V2cc (1/3)**
>
> We sincerely thank the reviewer for their insightful feedback & valuable suggestions. We address the reviewers' concerns below.
>
>
> **Real-world Generalizability:** To evaluate whether OWL relies on synthetic data and to examine its behavior in realistic acoustic conditions, we conducted additional experiments using the real-world DCASE Binaural Audio Scene Classification dataset [1] and the DCASE SELD Challenge 2021 dataset [2]. These datasets were selected because both are recorded in real environments. The first supports a general audio perception task, and the second supports a spatial localization task. Neither dataset is used during training, and both contain diverse environments that differ from our synthetic domains in microphone characteristics, background noise patterns, human activity, and reverberant structure.
>
> **OWL on Real-World Perception Task:** For zero-shot evaluation on Audio Scene Classification dataset, we used the following instruction-based prompt without any adaptation or fine-tuning:
>
> ```
> Based on the audio you have heard, refer to the instruction and provide a response.
> Your job is to predict the acoustic environment from the audio.
> Your answer must be one from the following environments:
> ['street pedestrian', 'metro', 'park', 'street traffic', 'metro station', 'public square', 'airport', 'shopping mall', 'bus', 'tram']
> ### Instruction:
> {instruction}
> ### Response:
> ```
>
> Table R1: OWL on binaural acoustic scene classification.
> | Class   | street pedestrian | metro | park | street traffic | Metro station | public square | airport | shopping mall | bus  | tram |
> |---------|-------------------|-------|------|----------------|---------------|---------------|---------|---------------|------|------|
> | Accuracy |              0.66 |  0.79 | 0.76 |           0.68 |          0.72 |          0.77 |    0.84 |          0.71 | 0.83 | 0.77 |
>
> The model achieves stable accuracy across all ten acoustic scenes (Table R1), including street pedestrian 0.66, metro 0.79, park 0.76, street traffic 0.68, metro station 0.72, public square 0.77, airport 0.84, shopping mall 0.71, bus 0.83, and tram 0.77. These results show that OWL generalizes well to real-world binaural recordings, despite being trained primarily on synthetic spatial audio. This behavior indicates that the geometry-aware audio encoder and the downstream reasoning components capture structure that transfers across simulation-to-real domain gaps.
>
> **OWL on Real-World Spatial Task:** To further demonstrate the compatibility of OWL with spatial reasoning tasks on real-world audio, we evaluate it on the DCASE SELD Challenge 2021 dataset. Because this dataset is recorded in first-order ambisonic format, we convert the FOA signals to binaural audio using an HRTF transformation [3]. This step is required because the SAGE encoder operates on binaural input and no real-world binaural spatial dataset is publicly available. Table R2 reports the performance of OWL on this dataset. OWL achieves consistent performance across both single-source and double-source conditions, which shows OWL's effectiveness in these real-world settings.
>
>
> Table R2: OWL on real-world spatial task.
>
> |   Method  	|          	|               	|   Type I             	|               	|
> |---------|---------------|-------------|-------------|-------------|
> |           	| Detection (mAP) 	|               	||   DoA (Acc)   	|               	|
> |           	|  Single Source  	| Double source 	| Single Source 	| Double source 	|
> | OWL       	|           57.21 	|         51.26 	|         42.78 	|         31.46 	|

---

> ### Author Response · Authors · 2025-11-19
> **Response to reviewer V2cc (2/3)**
>
> **CoT’s effect on Type I and Type II Questions:** Table 2 in the main paper refers to SELD performance of SAGE on BiDepth and SpaptialSoundQA datasets. CoT is not involved here. We believe the reviewer may be referring to Table 3, which reports OWL and OWL+CoT. In that table, the CoT variant does not substantially increase performance on Type I or Type II tasks, but rather maintains the performance of OWL’s without CoT version. For example, the single-source and double-source detection scores differ by only 0.06 and 0.02, and DoA accuracy differs by at most 0.07. These shifts do not indicate that CoT provides meaningful improvement on tasks that depend primarily on low-level spatial cues rather than multi-step reasoning. In short, CoT does not meaningfully improve Type I or Type II performance, but it maintains performance while enabling more structured outputs for downstream reasoning tasks (Types III and IV).
>
> **SAGE vs. General Audio Encoders:** The algorithm in [4] uses a Whisper encoder for ambisonic sound detection by combining the intensity vector with the omnidirectional channel. This approach is not directly comparable to SAGE because the intensity vector cannot be computed from binaural audio. Although their results are strong, they report performance only for left and right direction estimates, with accuracy that is comparable to SAGE. In future work, we plan to extend OWL to ambisonic data, which will allow a fair comparison with methods that rely on intensity based features.

---

> ### Author Response · Authors · 2025-11-19
> **Response to reviewer V2cc (3/3)**
>
> # References
>
> [1] Mars, Rohith, et al. "Acoustic scene classification from binaural signals using convolutional neural networks." (2019).
>
> [2] Politis, Archontis, et al. "A dataset of dynamic reverberant sound scenes with directional interferers for sound event localization and detection." arXiv preprint arXiv:2106.06999 (2021).
>
> [3]Noisternig, Markus. et al. "A 3D Ambisonic Based Binaural Sound Reproduction System." Presented at 24th AES International Conference: Multichannel Audio
>
> [4] Tang, C.,et al. Can Large Language Models Understand Spatial Audio? Proc. Interspeech 2024

---

> > ### Comment · Reviewer_V2cc · 2025-11-26
> >
> > Thanks for the rebuttal. The provided results and additional experiments have resolved my concerns. Please revise the paper to include these details. I have increased my rating.

---

> ### Author Response · Authors · 2025-11-24
> **Kind Reminder: Opportunity for Feedback on Our Submission**
>
> Dear Reviewer V2cc,
>
> As the discussion period is ongoing, we kindly wanted to check if you have any additional feedback or suggestions that we could address to further improve the paper. Please let us know if there’s anything we can clarify or revise that might positively impact your evaluation.
>
> Thank you for your time and thoughtful review.
>
> Regards,
>
> Authors of submission 3514

---

### Official Review · Reviewer_tQDg · 2025-10-31

**Soundness:** 4
**Presentation:** 4
**Contribution:** 4
**Rating:** 6
**Confidence:** 5

**Summary:**

The paper introduces SAGE (Spatial-Acoustic Geometry Encoder), a geometry-aware audio encoder, and OWL, an audio large language model (ALLM) that integrates SAGE with a chain-of-thought (CoT) spatial reasoning mechanism for auditory tasks. To support this approach, the authors curate BiDepth, a large-scale simulated dataset of over 1.1 million question-answer (QA) pairs, combining binaural audio, room impulse responses (RIR), panoramic depth images, and spatially grounded QA annotations. SAGE is supervised with paired geometric and acoustic data during training (but requires only audio at inference), and OWL builds on SAGE to enable structured spatial reasoning with interpretable rationales. Experiments across both new (BiDepth) and existing (SpatialSoundQA) datasets show consistent improvements in direction-of-arrival (DoA) estimation, spatial QA, and multi-step reasoning compared to prior work.

**Strengths:**

1. The introduction of SAGE, trained with auxiliary geometric supervision (via depth/RIR) but requiring only audio at inference, effectively bridges a key gap in current multimodal audio-language models (ALLMs), which often lack geometric grounding.

2. BiDepth represents a substantial, balanced, and diverse resource for training and evaluating geometry-aware audio-language systems.

3. Quantitative results demonstrate that SAGE and OWL outperform strong baselines (such as BAT and Spatial-AST) on both event detection and fine-grained DoA and spatial reasoning accuracy.

**Weaknesses:**

1. Reliance on synthetic data limits real-world generalizability.
2. The BiDepth dataset’s elevation coverage is heavily skewed toward the horizontal plane (Figure 5); this is realistic for indoor sources but may result in models struggling with uncommon vertical positions.

**Questions:**

1. Given the elevation bias displayed in Figure 5, how severe is OWL/SAGE’s performance degradation for sources at highly non-horizontal elevations?

---

> ### Author Response · Authors · 2025-11-19
> **Response to reviewer tQDg (1/3)**
>
> We sincerely thank the reviewer for their insightful feedback & valuable suggestions. We address the reviewers' concerns below.
>
>
> **Real-world Generalizability:** To evaluate whether OWL relies on synthetic data and to examine its behavior in realistic acoustic conditions, we conducted additional experiments using the real-world DCASE Binaural Audio Scene Classification dataset [1] and the DCASE SELD Challenge 2021 dataset [2]. These datasets were selected because both are recorded in real environments. The first supports a general audio perception task, and the second supports a spatial localization task. Neither dataset is used during training, and both contain diverse environments that differ from our synthetic domains in microphone characteristics, background noise patterns, human activity, and reverberant structure.
>
> **OWL on Real-World Perception Task:** For zero-shot evaluation on Audio Scene Classification dataset, we used the following instruction-based prompt without any adaptation or fine-tuning:
>
> ```
> Based on the audio you have heard, refer to the instruction and provide a response.
> Your job is to predict the acoustic environment from the audio.
> Your answer must be one from the following environments:
> ['street pedestrian', 'metro', 'park', 'street traffic', 'metro station', 'public square', 'airport', 'shopping mall', 'bus', 'tram']
> ### Instruction:
> {instruction}
> ### Response:
> ```
>
> Table R1: OWL on binaural acoustic scene classification.
> | Class   | street pedestrian | metro | park | street traffic | Metro station | public square | airport | shopping mall | bus  | tram |
> |---------|-------------------|-------|------|----------------|---------------|---------------|---------|---------------|------|------|
> | Accuracy |              0.66 |  0.79 | 0.76 |           0.68 |          0.72 |          0.77 |    0.84 |          0.71 | 0.83 | 0.77 |
>
> The model achieves stable accuracy across all ten acoustic scenes (Table R1), including street pedestrian 0.66, metro 0.79, park 0.76, street traffic 0.68, metro station 0.72, public square 0.77, airport 0.84, shopping mall 0.71, bus 0.83, and tram 0.77. These results show that OWL generalizes well to real-world binaural recordings, despite being trained primarily on synthetic spatial audio. This behavior indicates that the geometry-aware audio encoder and the downstream reasoning components capture structure that transfers across simulation-to-real domain gaps.
>
> **OWL on Real-World Spatial Task:** To further demonstrate the compatibility of OWL with spatial reasoning tasks on real-world audio, we evaluate it on the DCASE SELD Challenge 2021 dataset. Because this dataset is recorded in first-order ambisonic format, we convert the FOA signals to binaural audio using an HRTF transformation [3]. This step is required because the SAGE encoder operates on binaural input and no real-world binaural spatial dataset is publicly available. Table R2 reports the performance of OWL on this dataset. OWL achieves consistent performance across both single-source and double-source conditions, which shows OWL's effectiveness in these real-world settings.
>
>
> Table R2: OWL on real-world spatial task.
>
> |   Method  	|          	|               	|   Type I             	|               	|
> |---------|---------------|-------------|-------------|-------------|
> |           	| Detection (mAP) 	|               	||   DoA (Acc)   	|               	|
> |           	|  Single Source  	| Double source 	| Single Source 	| Double source 	|
> | OWL       	|           57.21 	|         51.26 	|         42.78 	|         31.46 	|

---

> ### Author Response · Authors · 2025-11-19
> **Response to reviewer tQDg (2/3)**
>
> **Azimuth Angle Distribution:** **BiDepth** is indeed simulated with indoor layouts in mind, and indoor acoustic events concentrate near the horizontal plane. We agree that this leads to a non-uniform elevation distribution. To evaluate whether this bias limits our models, we conducted additional experiments with an elevation distribution that follows a uniform distribution, contrasting with the original **BiDepth** configuration (Figure 12, Appendix H in updated manuscript).
>
>
>
> Table R3: **SAGE**'s performance on uniform elevation distribution
>
> |     Method    	| Modality 	|       	|BiDepth 	|       	(Azimuth with|       	 uniform |       	   Distribution)|
> |-------------|--------|-----|--------------------------------------------|-----|-----|-----|
> |               	|   Audio  	| Depth 	|                                        mAP                     	| $ER_{20^o}$ 	|  MAE  	|  DER  	|
> | SAGE$^*$ 	|  ✅      	|  ✅   	|                                        49.81 	| 28.13 	| 21.67 	| 14.32 	|
> | SAGE          	|  ✅      	| ❌    	|                                        49.74 	| 36.47 	| 26.22 	| 16.98 	|
> | SAGE          	|  ✅      	|  ✅   	|                                        49.82 	| 28.03 	| 21.58 	| 14.24 	|
>
> $^*$ Indicates with skewed elevation distribution
>
> The updated results in Table R3 show that SAGE performs consistently when evaluated under a uniform elevation distribution. mAP remains stable (49.80 vs. 49.81), and the error metrics ($ER_{20^o}$, MAE, DER) also remain consistent relative to the non-uniform elevation setting reported in Table 2 of the main paper. This suggests that SAGE does not rely on skewed elevation sampling and can effectively learn vertical localization cues even with uncommon vertical positions.
>
>
> Table R4: **OWL**'s performance on uniform elevation distribution
> |   Method  	|          	|               	|               Type I 	|               	|     Type II    	|               	|  Type III 	  |  	|             	|       	 Type IV  |
> |---------|---------------|-------------|-------------|-------------|--------------|-------------|--------|----------|--------|-----|
> |           	| Detection (mAP) 	|               	|   DoA (Acc)   	|               	| Distance (DER) 	|               	|          	|            	|          	|       	|
> |           	| Single Source 	| Double source 	| Single Source 	| Double source 	|  Single Source 	| Double source 	|    BA    	| Direction  	| Distance 	|          Avg  	|
> | OWL$^*$       	|           33.31 	|         17.24 	|         46.15 	|         34.25 	|          24.67 	|         31.29 	|    74.29 	| -          	| -        	| 65.27 	|
> | OWL + COT$^*$	|           33.37 	|         17.26 	|         46.17 	|         34.31 	|          23.29 	|         29.91 	|    77.89 	| 79.04      	| 86.76    	| 76.53 	|
> | OWL       	|           33.29 	|         17.28 	|         46.12 	|         33.12 	|          24.61 	|         31.46 	|    74.27 	| -          	| -        	| 65.19 	|
> | OWL + CoT 	|           33.39 	|         17.24 	|         46.14 	|         34.26 	|          23.17 	|         29.87 	|    77.56 	|      78.91 	|    86.68 	| 82.80 	|
>
> $^*$ Indicates with skewed elevation distribution
>
> We further evaluated OWL under the same uniform elevation distribution. Table R4 shows that OWL maintains strong performance across single-source and multi-source detection, DoA, and distance estimation tasks. The QA benchmarks (Type III and Type IV) remain comparable to the results in Table 3 of the main paper.

---

> ### Author Response · Authors · 2025-11-19
> **Response to reviewer tQDg (3/3)**
>
> # References
>
> [1] Mars, Rohith, et al. "Acoustic scene classification from binaural signals using convolutional neural networks." (2019).
>
> [2] Politis, Archontis, et al. "A dataset of dynamic reverberant sound scenes with directional interferers for sound event localization and detection." arXiv preprint arXiv:2106.06999 (2021).
>
> [3]Noisternig, Markus. et al. "A 3D Ambisonic Based Binaural Sound Reproduction System." Presented at 24th AES International Conference: Multichannel Audio

---

> ### Author Response · Authors · 2025-11-24
> **Kind Reminder: Opportunity for Feedback on Our Submission**
>
> Dear Reviewer tqDg,
>
> As the discussion period is ongoing, we kindly wanted to check if you have any additional feedback or suggestions that we could address to further improve the paper. Please let us know if there’s anything we can clarify or revise that might positively impact your evaluation.
>
> Thank you for your time and thoughtful review.
>
> Regards,
>
> Authors of submission 3514

---

### Official Review · Reviewer_2wCv · 2025-11-04

**Soundness:** 2
**Presentation:** 2
**Contribution:** 3
**Rating:** 4
**Confidence:** 4

**Summary:**

- This paper introduces OWL, an ALLM for geometry-aware spatial reasoning. It integrates an audio encoder with a spatially grounded CoT, enabling it to perform multi-step reasoning over direction and distance estimates. A large-scale dataset, coined BiDepth, is constructed to power this framework.
- The Spatial-Acoustic Geometry Encoder is trained to align binaural acoustic features with 3D spatial structure using panoramic depth images and simulated RIRs.
- Evaluated on BiDepth and the public SpatialSoundQA dataset, OWL demonstrates state-of-the-art SELD performance and spatial reasoning QA accuracy compared to major baselines.

**Strengths:**

- This work provides a comprehensive end-to-end framework for geometry-aware spatial reasoning, addressing all key components from the SAGE encoder and the OWL LLM to the specialized BiDepth training data.
- The paper introduces BiDepth, the first large-scale dataset of its kind, and its construction is sufficiently detailed in the appendix.
- The framework demonstrates significantly superior performance against both open-source and closed-source baselines, supported by extensive experimental validation and ablation studies.

**Weaknesses:**

- My main concern lies in the complexity and potential over-fitting of the proposed training pipeline. The framework requires a two-stage SAGE encoder pre-training, followed by a three-stage curriculum for the OWL model. While the ablation studies confirm this complex, multi-step process is necessary for the reported performance, it gives a strong impression of being tightly tailored to the custom-built BiDepth dataset. This raises significant questions about whether the approach is generally applicable or a highly specific solution.
- The contribution of using panoramic depth maps feels overstated, as this is a standard modality provided by the underlying SoundSpaces and Matterport simulation platforms. This is further confused by diagrams (Fig. 2) that imply the presence of specific sounding objects in depth maps (e.g., a cat), but the methodology does not discuss augmenting the 3D scenes with such new visual assets, only placing sound sources at coordinates.
- The design choice of dividing azimuth into 12 "o'clock" sectors feels ad-hoc and is not justified against other discretization schemes or as an established practice.

**Questions:**

Please refer to the weaknesses for detail. Almost all citations in the main draft are ill-formatted, making it quite uncomfortable to read.

---

> ### Author Response · Authors · 2025-11-19
> **Response to reviewer 2wCv (1/3)**
>
> We sincerely thank the reviewer for their insightful feedback & valuable suggestions. We address the reviewers' concerns below.
>
>
> We appreciate the reviewer’s concern regarding whether the multi-stage training pipeline is overly tailored to BiDepth. Our empirical results indicate that the approach exhibits strong generalization and is not dependent on dataset-specific artifacts.
>
> **Zero-shot transfer to SpatialSoundQA:** OWL also performs well on SpatialSoundQA without any fine-tuning. Zero-shot transfer provides strong evidence that the learned representations and reasoning components are not specialized to BiDepth. Table 2 of the main paper shows the zero-shot effectiveness of SAGE on unseen scenarios on the SELD task. Also, Table 4 in main paper demonstrates OWL’s generalizability on reasoning tasks on unseen scenarios.
>
> **Real-world Generalizability:** To evaluate whether OWL relies on synthetic data and to examine its behavior in realistic acoustic conditions, we conducted additional experiments using the real-world DCASE Binaural Audio Scene Classification dataset [1] and the DCASE SELD Challenge 2021 dataset [2]. These datasets were selected because both are recorded in real environments. The first supports a general audio perception task, and the second supports a spatial localization task. Neither dataset is used during training, and both contain diverse environments that differ from our synthetic domains in microphone characteristics, background noise patterns, human activity, and reverberant structure.
>
> **OWL on Real-World Perception Task:** For zero-shot evaluation on Audio Scene Classification dataset, we used the following instruction-based prompt without any adaptation or fine-tuning:
>
> ```
> Based on the audio you have heard, refer to the instruction and provide a response.
> Your job is to predict the acoustic environment from the audio.
> Your answer must be one from the following environments:
> ['street pedestrian', 'metro', 'park', 'street traffic', 'metro station', 'public square', 'airport', 'shopping mall', 'bus', 'tram']
> ### Instruction:
> {instruction}
> ### Response:
> ```
>
> Table R1: OWL on binaural acoustic scene classification.
> | Class   | street pedestrian | metro | park | street traffic | Metro station | public square | airport | shopping mall | bus  | tram |
> |---------|-------------------|-------|------|----------------|---------------|---------------|---------|---------------|------|------|
> | Accuracy |              0.66 |  0.79 | 0.76 |           0.68 |          0.72 |          0.77 |    0.84 |          0.71 | 0.83 | 0.77 |
>
> The model achieves stable accuracy across all ten acoustic scenes (Table R1), including street pedestrian 0.66, metro 0.79, park 0.76, street traffic 0.68, metro station 0.72, public square 0.77, airport 0.84, shopping mall 0.71, bus 0.83, and tram 0.77. These results show that OWL generalizes well to real-world binaural recordings, despite being trained primarily on synthetic spatial audio. This behavior indicates that the geometry-aware audio encoder and the downstream reasoning components capture structure that transfers across simulation-to-real domain gaps.
>
> **OWL on Real-World Spatial Task:** To further demonstrate the compatibility of OWL with spatial reasoning tasks on real-world audio, we evaluate it on the DCASE SELD Challenge 2021 dataset. Because this dataset is recorded in first-order ambisonic format, we convert the FOA signals to binaural audio using an HRTF transformation [3]. This step is required because the SAGE encoder operates on binaural input and no real-world binaural spatial dataset is publicly available. Table R2 reports the performance of OWL on this dataset. OWL achieves consistent performance across both single-source and double-source conditions, which shows OWL's effectiveness in these real-world settings.
>
> Table R2: OWL on real-world spatial task.
>
> |   Method  	|          	|               	|    Type I            	|               	|
> |---------------|---------------|-------------|-------------|-------------|
> |           	| Detection (mAP) 	|               	| |  DoA (Acc)   	|               	|
> |           	|  Single Source  	| Double source 	| Single Source 	| Double source 	|
> | OWL       	|           57.21 	|         51.26 	|         42.78 	|         31.46 	|
>
>
> ***These zero-shot results on unseen datasets (both synthetic and real-world) show that the model does not overfit during training and is not tailored to BiDepth alone.***

---

> ### Author Response · Authors · 2025-11-19
> **Response to reviewer 2wCv (2/3)**
>
> **Rationale for the training pipeline:** The multi-stage pipeline follows established practices in multimodal and audio-language modeling. Prior work commonly separates (i) modality encoder pre-training, (ii) alignment to the language backbone, and (iii) instruction or reasoning curriculum. Representative examples include BLIP and BLIP2 for vision-language training, LLaVA-style [3] alignment curriculum, and recent audio-LLMs such as Qwen-Audio[4], SALMONN [5], Audio Flamingo family [6-7], and Whisper-style encoder pre-training. Our two-stage SAGE encoder pre-training and three-stage OWL curriculum directly parallel these well-validated approaches while adapting them to spatial audio. The stages increase sample efficiency and stabilize optimization, which is critical when learning geometry-conditioned binaural cues.
>
> **Contribution of Panoramic Depth Image:** Panoramic depth is a central component of our approach because it provides global room geometry and occlusion structure that binaural audio alone cannot capture. Table 2 in the main paper shows that adding panoramic depth to SAGE improves all SELD metrics, especially on localization tasks e.g, (ER_20 decreases from 36.89 to 28.13; MAE decreases from 26.32 to 21.67). These gains demonstrate that depth is not redundant with audio and contributes measurable information.
> Regarding Figure 2, the cat was included only as a visual illustration to indicate the origin of the sound source. The simulation does not insert new objects but places sound emitters at coordinates in existing scenes. To avoid confusion, we have revised Figure 2 to remove this ambiguity.
>
> **Design Choice of Dividing Azimuth into 12 o’clock Sectors:** Our choice of 12 clockface sectors provides a balance between expressiveness and human interpretability. A full 360 degree discretization offers a level of precision that users cannot reliably judge in natural listening conditions, while a coarse four-quadrant scheme lacks sufficient directional detail. The 12-sector clockface is a long-standing representation [8] in auditory spatial descriptions and gives more precision than simple front or side labels while remaining practical and intuitive for both annotation and model evaluation.
>
>
>
> **Citation Formatting:** The citation style follows the official ICLR template, which controls the formatting. We will review our BibTeX entries to ensure they compile cleanly within the template.

---

> ### Author Response · Authors · 2025-11-19
> **Response to reviewer 2wCv (3/3)**
>
> # References
> [1] Mars, Rohith, et al. "Acoustic scene classification from binaural signals using convolutional neural networks." (2019).
>
> [2] Politis, Archontis, et al. "A dataset of dynamic reverberant sound scenes with directional interferers for sound event localization and detection." arXiv preprint arXiv:2106.06999 (2021).
>
> [3] Liu, Haotian, et al. "Visual instruction tuning." Advances in neural information processing systems 36 (2023): 34892-34916.
>
> [4] Chu, Yunfei, et al. "Qwen-audio: Advancing universal audio understanding via unified large-scale audio-language models." arXiv preprint arXiv:2311.07919 (2023).
>
> [5] Tang, Changli, et al. "Salmonn: Towards generic hearing abilities for large language models." arXiv preprint arXiv:2310.13289 (2023).
>
> [6] Kong, Zhifeng, et al. "Audio flamingo: A novel audio language model with few-shot learning and dialogue abilities." arXiv preprint arXiv:2402.01831 (2024).
>
> [7] Ghosh, Sreyan, et al. "Audio flamingo 2: An audio-language model with long-audio understanding and expert reasoning abilities." arXiv preprint arXiv:2503.03983 (2025).
>
> [8]Ye, Peng, et al. "Positioning localities for vague spatial location description: A supervaluation semantics approach." ISPRS International Journal of Geo-Information 11.1 (2022): 68.

---

> ### Author Response · Authors · 2025-11-24
> **Kind Reminder: Opportunity for Feedback on Our Submission**
>
> Dear Reviewer 2wCv,
>
> As the discussion period is ongoing, we kindly wanted to check if you have any additional feedback or suggestions that we could address to further improve the paper. Please let us know if there’s anything we can clarify or revise that might positively impact your evaluation.
>
> Thank you for your time and thoughtful review.
>
> Regards,
>
> Authors of submission 3514

---

### Author Response · Authors · 2025-11-29
**Summary of reviews and author responses for AC**

We thank all reviewers for their constructive feedback and for recognizing the strengths and potential impact of this work in advancing spatial audio reasoning. Their comments helped us improve the clarity and completeness of the paper, and we appreciate the careful attention given to our methodology, dataset, and experimental validation. We also thank the Area Chair for their time and thoughtful consideration in evaluating our submission.


> ### Strengths acknowledged consistently across reviews

1. Reviewers recognize the **BiDepth** dataset as a substantial contribution. Its scale, structure, and focus on spatial audio make it *a valuable resource that can support future research in spatial perception and reasoning.*
2. Reviewers highlight the **technical novelty of the proposed geometry-aware audio encoder, SAGE**. They note the clear rationale behind the design and training choices, including the use of auxiliary geometric signals during training while requiring only audio at inference. *They also agree that the method addresses an important gap in current audio-language models, which typically lack geometric grounding.*
3. Reviewers acknowledge the **breadth and rigor of the empirical evaluation**. The paper includes extensive ablations, comparisons to strong baselines, and results across public datasets for both sound event detection and localization, as well as broader spatial reasoning tasks.


> ### Concerns raised by the reviewers and our responses


### Real-world generalizability (reviewers tQDg and V2cc).
To address concerns about reliance on synthetic data, we added experiments on two real-world datasets: the DCASE Binaural Audio Scene Classification dataset and the DCASE SELD Challenge 2021 dataset. Both are recorded in real environments and cover general audio perception as well as spatial localization tasks. These results confirm the generalizability of  **OWL** and **SAGE** on real-world scenarios. *Please refer to [Response to reviewer V2cc (1/3)](https://openreview.net/forum?id=zPv46YKv3w&noteId=jV7duzHpFX) for detailed results.*

### Performance on uncommon vertical angles (reviewer tQDg).
We conducted additional experiments using an elevation distribution that follows a uniform configuration rather than the original BiDepth distribution. The results demonstrate that the model adapts robustly to uncommon elevation angles. *Details are provided in the [Response to reviewer tQDg (2/3)](https://openreview.net/forum?id=zPv46YKv3w&noteId=K02L7nbtx6)*.

### Potential overfitting due to training methodology (reviewer 2wCv).
Zero-shot evaluations on multiple unseen datasets, including both synthetic and real-world sources, show that the model does not overfit the training distribution and is not specialized to BiDepth. *The [Response to reviewer 2wCv (1/3)](https://openreview.net/forum?id=zPv46YKv3w&noteId=oP4tymZxAg) provides detailed results.*

### Contribution of panoramic depth (reviewer 2wCv).
Panoramic depth provides global room geometry and occlusion structure that binaural audio alone cannot supply. *Table 2 in the main paper* shows consistent improvements when depth is added, particularly for localization accuracy. For example, $ER_{20^o}$ improves from 36.89 to 28.13, and MAE improves from 26.32 to 21.67. These gains confirm that depth information provides non-redundant and measurable value.


We also addressed all **other clarification questions** raised by the reviewers in our detailed responses to their thoughtful comments.

***Reviewer V2cc confirmed that these experiments and clarifications resolved their concern and raised their score from 6 to 8 before the discussion period ended prematurely.***

We hope that our clarifications and additional experiments address the reviewers' concerns and demonstrate the robustness and significance of our contributions. We appreciate the opportunity to respond and thank the Area Chair once again for their careful evaluation and consideration of our work.

---

### Meta-Review · Area_Chair_5okA · 2026-01-07

**Summary:**

This paper received mixed reviews. The main concerns raised in the reviews are:
1. complexity of the training pipeline, which may hurt generality (`2wCv`).
2. over-claim of the contribution of panoramic depth maps (`2wCv`).
3. no evaluation on real-world scenes (`tQDg`, `V2cc`).
4. no evaluation with varying vertical positions (`tQDg`).

Overall, the reviewers do find merits in the paper. The main complaint of missing real-world evaluation has largely been addressed by the new results. A few minor concerns are still outstanding, but I believe these can be fixed in the final revision. The submission should be ready for publication.

One additional note: I don't think it is appropriate to call OWL itself an "audio large language model (ALLM)". An ALLM suggests more generic capabilities than spatial reasoning.

**Reviewer Concerns:**

1. Concern #1 is partially addressed. The generalization of the method is validated by the new results on real-world scenes. The rebuttal did not directly address the concern on the complexity of the method.
2. Concern #2 has not been directly addressed in the rebuttal.
3. Concern #3 is largely addressed by the new results on real-world datasets, although no baseline comparisons were presented in these settings.
4. Concern #4 is largely addressed by the new results with a diverse elevation distribution, which show little difference in metrics.

**Reviewer Scores:**

1. Reviewer `2wCv` (4->4+?): the reviewer may or may not increase the rating as some of the concerns have not been fully addressed.
2. Reviewer `tQDg` (6->6+): the reviewer might increase the rating given the new results on real-world scenes and varying vertical positions, directly addressing the two concerns raised.
3. Reviewer `V2cc` (6->8): the reviewer has indicated that they would increase the rating.

---

### Decision · Program_Chairs · 2026-01-26

Accept (Poster)